# Validating the concept of mutational signatures with isogenic cell models

Xueqing Zou[1], Michel Owusu[2], Rebecca Harris[1], Stephen P. Jackson[3], Joanna I. Loizou[2] & Serena Nik-Zainal[1,4]

The diversity of somatic mutations in human cancers can be decomposed into individual mutational signatures, patterns of mutagenesis that arise because of DNA damage and DNA repair processes that have occurred in cells as they evolved towards malignancy. Correlations between mutational signatures and environmental exposures, enzymatic activities and genetic defects have been described, but human cancers are not ideal experimental systems —the exposures to different mutational processes in a patient's lifetime are uncontrolled and any relationships observed can only be described as an association. Here, we demonstrate the proof-of-principle that it is possible to recreate cancer mutational signatures in vitro using CRISPR-Cas9-based gene-editing experiments in an isogenic human-cell system. We provide experimental and algorithmic methods to discover mutational signatures generated under highly experimentally-controlled conditions. Our in vitro findings strikingly recapitulate in vivo observations of cancer data, fundamentally validating the concept of (particularly) endogenously-arising mutational signatures.

[1] Wellcome Trust Sanger Institute, Wellcome Genome Campus, Hinxton CB10 1SA, UK. [2] CeMM Research Center for Molecular Medicine of the Austrian Academy of Sciences, Lazarettgasse 14, AKH BT 25.3, 1090 Vienna, Austria. [3] The Gurdon Institute and Department of Biochemistry, University of Cambridge, Cambridge CB2 1QN, UK. [4] Department of Medical Genetics, The Clinical School, University of Cambridge, Cambridge CB2 0QQ, UK. These authors contributed equally: Xueqing Zou, Michel Owusu. Correspondence and requests for materials should be addressed to J.I.L. (email: jloizou@cemm.oeaw.ac.at) or to S.N.-Z. (email: snz@sanger.ac.uk)

The concept of mutational signatures was postulated in 2012: The catalogue of somatic mutations uncovered through tumour sequencing is the outcome of one or more mutational processes that have been operative through the lifetime of a cancer patient[1,2]. Each mutational process, defined by DNA damage and DNA repair components, leaves a characteristic pattern or *mutational signature* on the tumour genome[1–4]. The final mutational portrait of each patient's cancer is determined by the intensity and duration of exposure to each mutational process[4,5].

As an analytical principle, mutational signatures have gained considerable traction, and are regularly featured in cancer genomics literature[6–8]. Already, there are multiple algorithms to extract mutational signatures[5,9–12], though each has its own mathematical idiosyncrasies leading to results that are broadly similar, but never identical. This has caused some to question the robustness of the concept. Nevertheless, as a field, mutational signature research has progressed remarkably. Mutational signatures have been sought across tens of thousands of cancers, revealing over 40 different base substitution signatures (paper in preparation), further supplemented by assessments of how these signatures are distributed across various genomic architectures including replication-timing domains, replication strands, nucleosome occupancy and transcription factor binding sites[13,14]. More recently, genome rearrangement signatures have been unveiled, assisting in the categorization of breast cancer subtypes[13,15,16] and clinical applications based on mutational signatures are currently being developed[17].

No matter how sophisticated the analyses of in vivo mutagenesis of cancers, there are limitations to studying tumours—it is an uncontrolled and noisy system[18–21], and even the best clinical metadata collections will at most, provide associations. Critics of the concept have highlighted that this purely mathematically-based idea, although compelling, lacks definitive validation through in vitro methods.

Historic *TP53* and *HPRT* reporter assays and experiments exposing mouse embryonic fibroblasts (MEFs) to various exogenous agents have already provided convincing evidence that mutation patterns can be generated, particularly for environmental agents such as ultraviolet light and tobacco carcinogens[22,23]. Yet, there have been limited efforts to demonstrate similarly clear relationships for endogenous mutational processes. Few would dispute that substitution Signature 1 composed primarily of C>T transitions at an NpCpG sequence context is linked with deamination of methyl-cytosines, and substitution Signatures 2 and 13 characterised by the distinctive C>T transitions and C > G transversions at a TpCpN trinucleotide context

are initiated by the activity of the APOBEC family of enzymes[3,4]. However, many of the mutational signatures that are likely to be endogenous in origin have not been verified. Associations of specific substitution and insertion/deletion (indel) signatures with mismatch repair (MMR) deficiency[24–26], as well as substitution, indel and rearrangement signatures with homologous recombinational (HR) repair deficiency[27–30] though conspicuous, have not been confirmed. Many other genes are also involved in the myriad DNA repair pathways in our cells, and it is not clear whether genetic defects in alternative, related genes could produce mutational signatures as well. Even if mutational signatures could be reproduced using in vitro techniques, it is not known whether these signatures would mimic what is observed in vivo.

Here, we explore whether targeted CRISPR-Cas9-based[31–33] knockouts of selected DNA repair genes can recreate mutational signatures. We describe the experimental cell-based system and develop the computational methodologies to confirm or refute whether each gene knockout generates mutation patterns, thus, providing a general approach for exploring mutational signatures. We further seek whether experimentally-generated mutation patterns bear similar appearances and/or behaviours to mutational signatures seen in primary cancers. If so, this would serve to endorse that mutational signatures are not simply mathematical extractions, but are the consequences of true biological processes.

## Results

**Generation of DNA repair gene knockouts.** We used the immortalised human near-haploid cell line HAP1 to generate isogenic CRISPR-Cas9-mediated knockouts[34]. The advantage of using a haploid cell line is that CRISPR-Cas9-mediated editing is simplified because only one genetic allele needs to be altered to generate a null phenotype. Moreover, because only half the genomic DNA is present, next generation sequencing (NGS) needs are substantially reduced making the experiment more affordable. To determine whether we could detect mutational signatures that result from defects in DNA repair pathways we chose to target genes that play diverse and independent roles in the detection, signalling or repair of DNA damage (Table 1).

Aliquots of the HAP1 cell line were exposed to constructs that express the endonuclease Cas9 and guide RNAs (gRNAs) that were designed to target individual genes of interest. Single clones were selected and those carrying a frame-shift mutation in the given gene were designated as the parental cell line (Fig. 1a), which were amplified and analysed by high-depth whole genome

**Table 1 List of DNA repair genes targeted and their functions**

| Gene symbol | Gene name | Function | Repair pathway | Position |
|---|---|---|---|---|
| CHK2 | Checkpoint kinase 2 | Serine threonine kinase | Cell cycle and apoptotic regulation in response to DNA damage | 22q12.1 |
| EXO1 | Exonuclease 1 | 5′ to 3′ exonuclease; RNase H activity | Homologous recombination; mismatch repair | 1q43 |
| FANCC | Fanconi anemia, Complementation group C | Component of Fanconi repair system core complex | DNA cross-link repair | 9q22.32 |
| MSH6 | MutS homolog 6 | Mismatch recognition | Mismatch repair | 2p16.3 |
| NEIL1 | Endonuclease VIII-like 1 | DNA glycosylase and apurinic/apyrimidinic lyase | Base excision repair | 15q24.2 |
| NUDT1 | Nudix hydrolase 1 | Hydrolyzes oxidized purine nucleoside triphosphates | Modulation of nucleotide pools | 7p22.3 |
| POLB | DNA polymerase beta | DNA polymerase (catalytic subunit) | Base excision repair | 8p11.21 |
| POLE | DNA polymerase epsilon | DNA polymerase (catalytic subunit) | Nucleotide excision repair and mismatch repair | 12q24.33 |
| POLM | DNA polymerase mu | DNA polymerase (catalytic subunit) | Gap filling during non-homologous end-joining | 7p13 |

sequencing (WGS). The parental cell lines (labelled as 'parental clone' in Fig. 1a) were subsequently cultured for one month, from which seven 'subclones' were derived, amplified and analysed by WGS. This workflow served to allow for the identification of mutations that occurred over approximately 36 cellular divisions, considering that the doubling time is approximately 20 h.

Each parental clone and subclone was successfully sequenced to ~15-fold depth. Short read sequences were aligned to the human reference genome assembly GRCh37/hg19 and all classes of somatic mutations were called in the parental clones (subtracting from the primary bulk HAP1 population) and in subclones (subtracting from the parental clones). Targeting of the genes of interest was confirmed by identifying frameshift indels in the relevant gene in short-read data (see Supplementary Fig. 1a and Supplementary Data 1), and loss of protein expression was confirmed through immunoblotting (Supplementary Fig. 1b). Potential off-target edits were also systematically sought in an agnostic manner, whether generating small or large (multi-kb) insertion or deletions, and none were identified. Proliferation rates were also determined for each knockout cell line (Supplementary Fig. 1c). Moreover, potential off-target sites were also searched using COSMID (http://crispr.bme.gatech.edu), a web-based tool to identify and validate CRISPR/CAS9 off-target sites[35] (see Supplementary Data 2 for a ranked-list of potential off-target sites of the relevant guide RNA sequences generated by COSMID). Furthermore, we also confirmed in all subclones, that no additional mutations were acquired in other DNA repair genes during the early clonal expansion phase (see Supplementary Data 3 for a list of DNA repair genes) that could affect the final mutational signature obtained in each subclone.

**Knockouts of DNA repair genes instigates mutagenesis**. A level of background mutagenesis was observed in parental clones (average ~1200 substitutions, ~60 indels, ~6 rearrangements) and in all subclones (Fig. 1b–d and Supplementary Figs. 2–4). Above the background mutations, subclones associated with particular gene knockouts also had greater numbers of specific classes of mutations, although effect sizes were notably variable. For example, the knockout of MSH6 was associated with a surge of substitutions and indels. By contrast, the FANCC knockout was associated with a possibly small increase in indels but a large increase in rearrangements. Knockout of EXO1 appeared to cause modest elevations of all classes of mutation (Fig. 1b–d). For each gene knockout, a high level of consistency was observed between all seven subclones in terms of total counts (Fig. 1b–d) and overall patterns of mutations (Supplementary Figs. 2–4). Thus, at first pass, it is possible to crudely discriminate between the effects of gene knockouts through these experiments, suggesting that this is a rational experimental system for exploring the mutational effects conferred by defects in specific genes.

**Understanding the signal-to-noise issue**. There are however a number of issues to acknowledge and resolve which are universal to all human cell-based systems used for exploring mutagenesis. First, the background mutagenesis was easily detectable: for example, for base substitutions approximately 700–2000 mutations were detected per colony and this comprises a distinctive C>A/G>T substitution pattern with tallest peaks at TCT, GCA, GCT and ACA (in decreasing order; Supplementary Fig. 2). This ubiquitous signature shares considerable similarity with previously reported Signature 18, first observed in primary neuroblastoma[3]. Subsequently, this mutational signature was described in breast and adrenocortical cancers. A very similar signature (cosine similarity of 0.94 to Signature 18) has been associated with mutations in the MUTYH gene, hinting that it is a final

outcome of a primary mutational process that could involve oxidative damage[8]. Regardless, this mutational process was effectively noise in our system, and was pervasive in parental clones and subclones in our experiments, supporting the possibility of it being due to DNA damage incurred during the experimental process. Background mutagenesis was also detectable in indels (Supplementary Fig. 3) and rearrangements (Supplementary Fig. 4).

Second, this inescapable and abundant mutational process contributed a very large volume of background mutagenesis, which could complicate the detection of true mutational signatures for each target knockout gene. The mutation signals of various gene knockouts were highly different—some were strong in nature while others may be considerably weaker, and could be obscured by the overwhelming background signature. These two issues of high noise and potentially low signal are generic and arise in other cell-based models including induced pluripotent stem cells (iPSCs)[36], embryonic stem cells (ESCs) (manuscript in preparation) and organoids[36–38]. As described below, we thus developed methods to quantitatively and reliably discern whether mutational signatures are present in cell-based experimental systems in order that they may be applied to similar approaches in the future.

**Detecting mutational signatures in experimental systems**. The pervasive background signature was present in all parental clones and subclones regardless of gene knockout. By contrast, if a gene knockout produced a mutational signature, then the signature should be observed in all relevant subclones and would not be detectable (or be present at a greatly reduced level) in the parental clone. We do however, expect some variation between subclones and must therefore take this into consideration in the modelling. Our aim therefore was to determine whether there is robust and consistent divergence of subclones from parental clones, both qualitatively (mutation spectrum) and quantitatively (mutation count), indicative that targeting particular genes does indeed produce mutational signatures.

To account for the limited number of samples and mutations per sample, and the potentially limited signal-to-noise ratio, we used a bootstrap resampling method of the 96-channel mutation profile for all parental clones and subclones (Fig. 2 and Online Methods for details). This provided us with distributions of subclones and of parental clones from which reliable estimates of the qualitative differences in mutation spectra could be calculated (Fig. 2a; see Online Methods for details). An additional tier to discriminate whether a gene knockout is associated with mutagenesis came from taking mutation count into consideration: an "expected" mutation density was used to deduce a $p$ value to detect an alteration in mutation burden for subclones of a given gene knockout (Fig. 2b, see Online Methods for details). Once a gene knockout was confirmed to be associated with generating a mutation pattern, the final mutational profile (which is a linear combination of background mutagenesis and the gene knockout) was obtained by subtracting the background mutagenesis from the mutational profile of the subclones (see Online Methods).

This principle of signature discrimination (Fig. 2c) was applied to indel and rearrangement patterns as well, although different classifications were used. For indels, a vector of eight features was used comprising the following categories: 1 bp insertion, >=2 bp insertion, 2 bp microhomology-mediated deletion, >= 3 bp microhomology-mediated deletion, 1 bp repeat-mediated deletion, >=2 bp repeat-mediated deletion, other deletion (where there are no specific junctional features associated) and complex indels. For rearrangements, a vector of ten features was applied

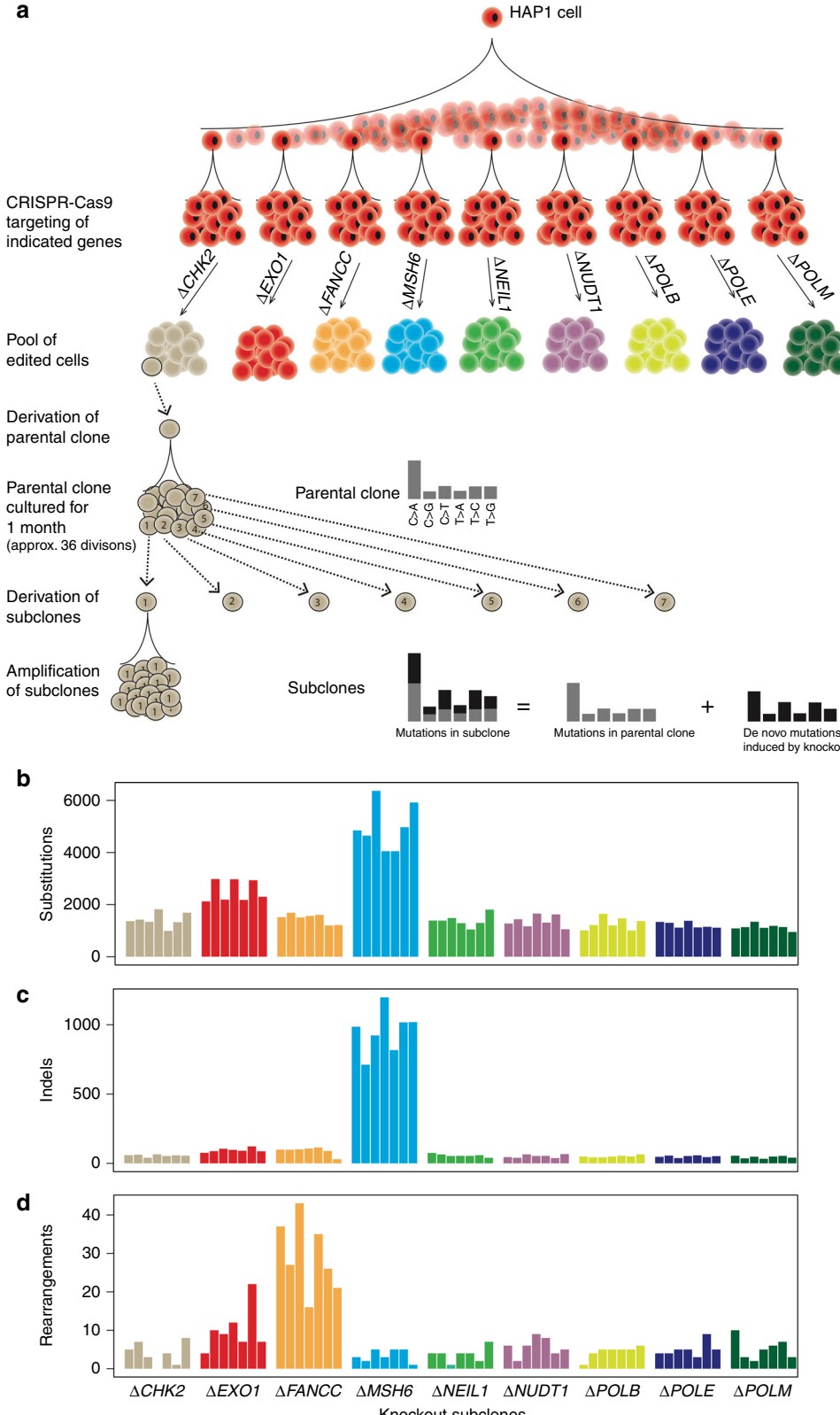

**Fig. 1** Knockouts of DNA repair genes instigate mutagenesis. **a** The experimental strategy for investigating whether DNA repair gene knockouts produced mutagenic effects. Parental HAP1 cells are split into multiple aliquots and used for CRISPR-Cas9-mediated gene-editing of the indicated genes. Resulting clones carrying frame-shift mutations are identified by Sanger sequencing and immunoblotting, amplified, cultured for one month (approximately 36 divisions), and seven subclones are derived through a single-cell bottleneck. DNA is extracted and whole genome sequenced for the seven subclones, and the parental clone. De novo mutations in a subclone that is subject to a particular knockout can be obtained by removing mutations in the parental clone from mutations in the subclone. De novo mutations are identified for all classes of mutation: **b** substitutions, **c** indels and **d** rearrangements, of the seven subclones for each knockout gene

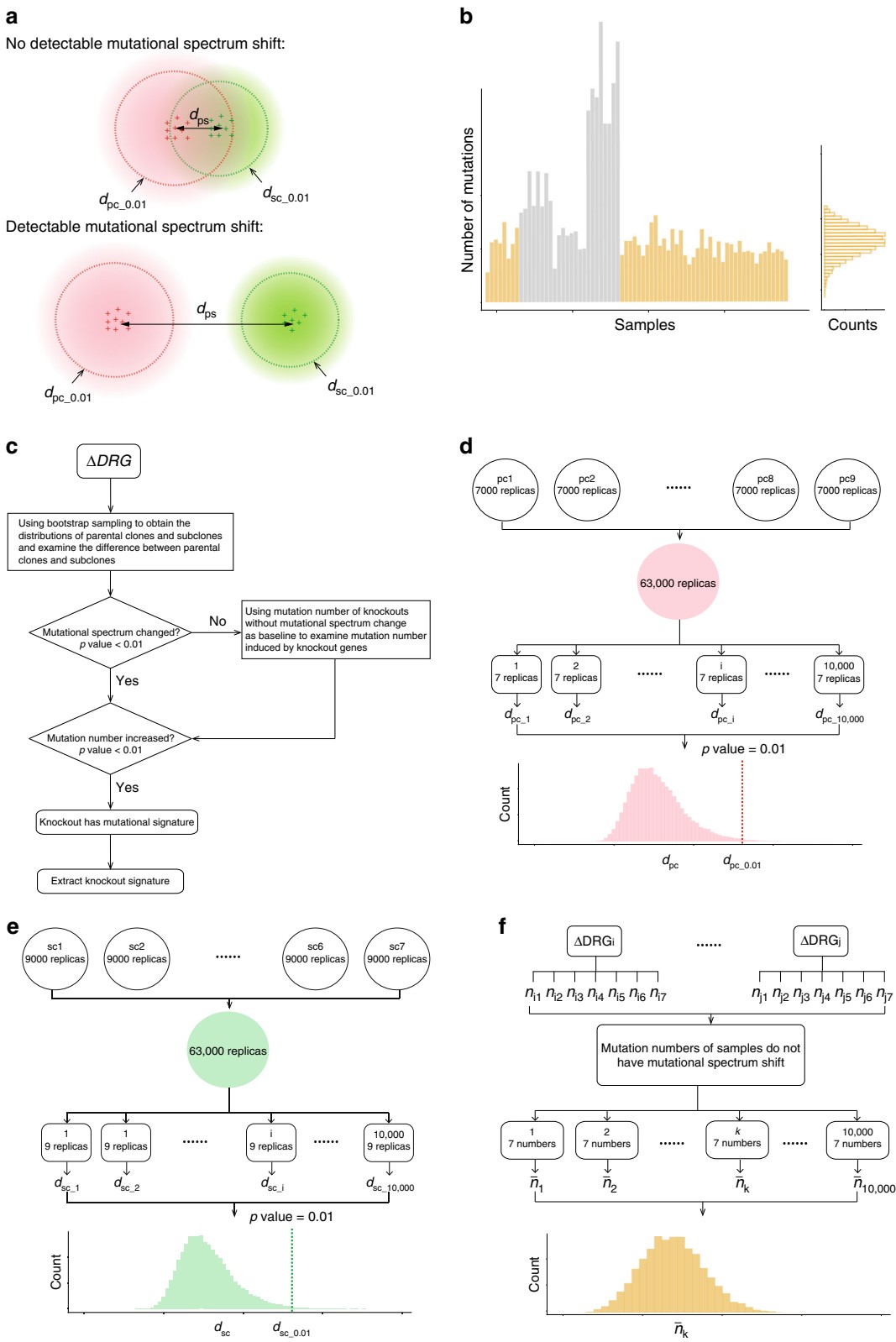

comprising: 1–10 kb, 10 kb–1 Mb and >1 Mb size groups of the three classes of deletions, inversions and tandem-duplications, and the last category was translocations.

By using these methods, we conclusively identified seven mutational signatures from nine gene knockouts in this HAP1-based experimental system: two substitution signatures were induced by knockouts of *EXO1* and *MSH6* (Fig. 3); three indel signatures produced by knockouts of *EXO1*, *FANCC* and *MSH6* (Fig. 4); and two rearrangement signatures associated with knockouts of *EXO1* and *FANCC* (Fig. 5), as described in detail below.

**Experimentally-generated gene knockout mutational signatures**. MSH6 is a protein involved in DNA MMR. MSH6 forms a heterodimer with MSH2 and helps to maintain a low error rate during replication[39]. Inherited mutations in this gene are associated with elevated risks particularly of colorectal and endometrial cancer[40,41]. Inherited and somatic mutations with loss of the wild-type allele are associated with elevated mutation rates in primary human cancers, particularly at polynucleotide repeat tracts conferring a diagnostic phenotype called micro-satellite instability (MSI)[26]. In-keeping with previous observations, the MSH6 knockout was associated with considerably elevated substitution density (~4 fold) over background and had a characteristic pattern dominated by C>T and T>C mutations (Fig. 3a, d). This mutational signature bears a resemblance to the multiple substitution signatures (extracted from many different tumour-types) that have been associated with MMR deficiency in cancers (Signatures 6, 12, 14, 15, 20 and 26), but was not perfectly identical to any one of them. Interestingly, when mutational signatures are extracted from breast cancers alone and all analyses restricted to just this tissue-type, we find that the in vitro signature is strikingly similar to the MMR deficiency signature in breast cancers. This is also the case for tumour-specific signature extractions of 52 colorectal and 44 endometrial cancers, both being cancer-types that are associated with MSH6 mutations. Furthermore, the MSH6 knockout had a very high level of 1 bp deletions occurring at polynucleotide repeat tracts, with ~7 fold more deletions than insertions overall, in-keeping with MSI (Fig. 4a, d). Intriguingly, an MSH6 knockout in an alternative iPSC model generated an identical signature (cosine similarity is 0.94) suggesting that in different cell lines, the signature associated with MSH6 knockout is very stable (unpublished data).

EXO1 encodes an enzyme that functions as a 5′-3′ DNA exonuclease as well as an endonuclease cleaving RNA on DNA/RNA hybrids (RNase H activity)[42–44]. It plays a role in, and interacts with, components of both the DNA double-strand break repair (DSBR) and MMR[45] pathways. The EXO1 knockout resulted in a substitution signature with predominantly C>A/G>T transversions with peaks at GCT, GCC and TCT (Fig. 3d) and smaller contributions from C>G/G>C, C>T/G>A and T>C/A>G. The EXO1 knockout also had an indel pattern that featured a high percentage of 1 bp repeat-mediated deletions and a smaller proportion of long (>=3 bp) microhomology-mediated deletions (mm-del) (Fig. 4d). This is an example where the indel knockout signature and background signature are qualitatively similar (cosine similarity is 0.97, Fig. 4b) but quantitatively distinct (Fig. 4c). Additionally, the EXO1 knockout produced a rearrangement signature characterised predominantly by a high percentage (60%) of medium-to-large (10 Kb–1 Mb) tandem duplications (Fig. 5d). Knockout of EXO1 thus created multiple signatures of all mutation classes, probably as a consequence of EXO1 operating at the junction of several DNA repair pathways.

FANCC is a component of the Fanconi anemia (FA) DNA repair system that functions in the processing of DNA crosslinks that are encountered in S phase via a mechanism that ultimately employs homologous recombination (HR)[28,46,47]. In-keeping with this role, the FANCC knockout created a number of mutational signatures that are predicted to be initiated by a DNA double-strand break. These included a characteristic indel pattern of long deletions (⩾3 bp in length) with microhomology observed at the indel junction (Fig. 4d). Furthermore, the FANCC knockout produced a rearrangement pattern characterised by chromosomal deletions of between 1–10 Kb in size, inversions in all size ranges, as well as short (=<10 Kb) and long (>1 Mb) tandem duplications (Fig. 5d). This combination of indel and rearrangement patterns showed a high degree of similarity to those seen in primary tumours with defects of other well-known HR components such as BRCA1 and BRCA2[15,17].

To understand whether the targeting of these DNA repair genes could affect proliferation, we measured the proliferation rates of the given cell lines over a period of ten days (Supplementary Fig. 1c). The MSH6, EXO1 and FANCC knockouts had the slowest proliferation rate, indicating that loss of these genes is not associated with an increased proliferative rate. Hence, the elevated numbers of mutations in MSH6, EXO1 and FANCC knockouts were not simply due to an increase in the rate of cell division. Based on these assays, the mutation rates of the seven mutational signatures can be calculated: MSH6 knockout signatures produced ~148 substitutions and ~36 indels per cell division; EXO1 knockout signatures produced ~16 substitutions, ~0.58 indels and ~0.19 rearrangements per cell division; FANCC knockout signatures produced ~0.58 indels and ~0.68 rearrangements per cell division (Supplementary Data 4).

The knockouts of CHK2[48–50], NEIL1[51], NUDT1[52], POLB[53], POLE[54] and POLM[55] did not appear to produce detectable mutational signatures under these experimental conditions. Additionally, apart from the gene-edits themselves, there were no additional recurrent activating mutations or loss-of-function mutations identified in subclones after culture, suggesting that the enrichment of "driver" events was not a feature in these experiments.

Somatic mutations in DNA polymerase epsilon (POLE) have been reported to be associated with a characteristic mutational process in Signature 10[56,57]. We found however that knockout of POLE, did not appear to be associated with a striking signature in our study. This is not surprising, given that the identified mutational signature is associated with mutations in the proof-reading domain of POLE (dominant negative effect), which is not mimicked by the knockout.

These results highlight successful, methodically-generated genome-wide mutation patterns of all classes, in a human cell-based system, demonstrating that biological abrogation of some DNA repair genes not only initiates mutagenesis, but

**Fig. 2** Schematic illustration of algorithm developed in the present study. **a** Schematic illustration distinguishing the mutational spectrum of parental clones and subclones. Each red "+" represents a parent clone and green "+" represents a subclone. Red and green clouds represent bootstrapped samples for parental clones and subclones respectively. $d_{ps}$ is the distance between the centroid of parental clones and that of subclones. Red dashed circle shows the boundary of distance $d_{pc}$ with $p$ value = 0.01 and green dashed circle shows the boundary of distance $d_{sc}$ with $p$ value = 0.01 (see online Methods). The mutational spectra of parental clones and subclones are considered to be different only when $d_{ps} > d_{pc\_0.01}$ and $d_{ps} > d_{sc\_0.01}$. **b** Distribution of background mutation number in subclones. Left: The number of mutations in each sample. Cyber yellow and grey highlight the samples that do not have or do have mutational spectrum shifts from parental clones, respectively. Right: Mutation numbers of the samples that do not have mutational spectrum shifts (cyber yellow samples) are used to construct a distribution indicating expected numbers of mutations in cells where the gene knockout does not have an effect. **c** Workflow of characterisation on knockout signatures. **d**, **e** Detailed workflow of quantitative estimation of the difference between the mutation spectrum of parental clones and that of subclones by bootstrapping parental clones (**d**) and subclones (**e**) (see Online Methods). **f** Detailed work flow of the construction of the distribution of mutation numbers generated in cells where the gene knockout does not have an effect, using bootstrap sampling methods (see Online Methods)

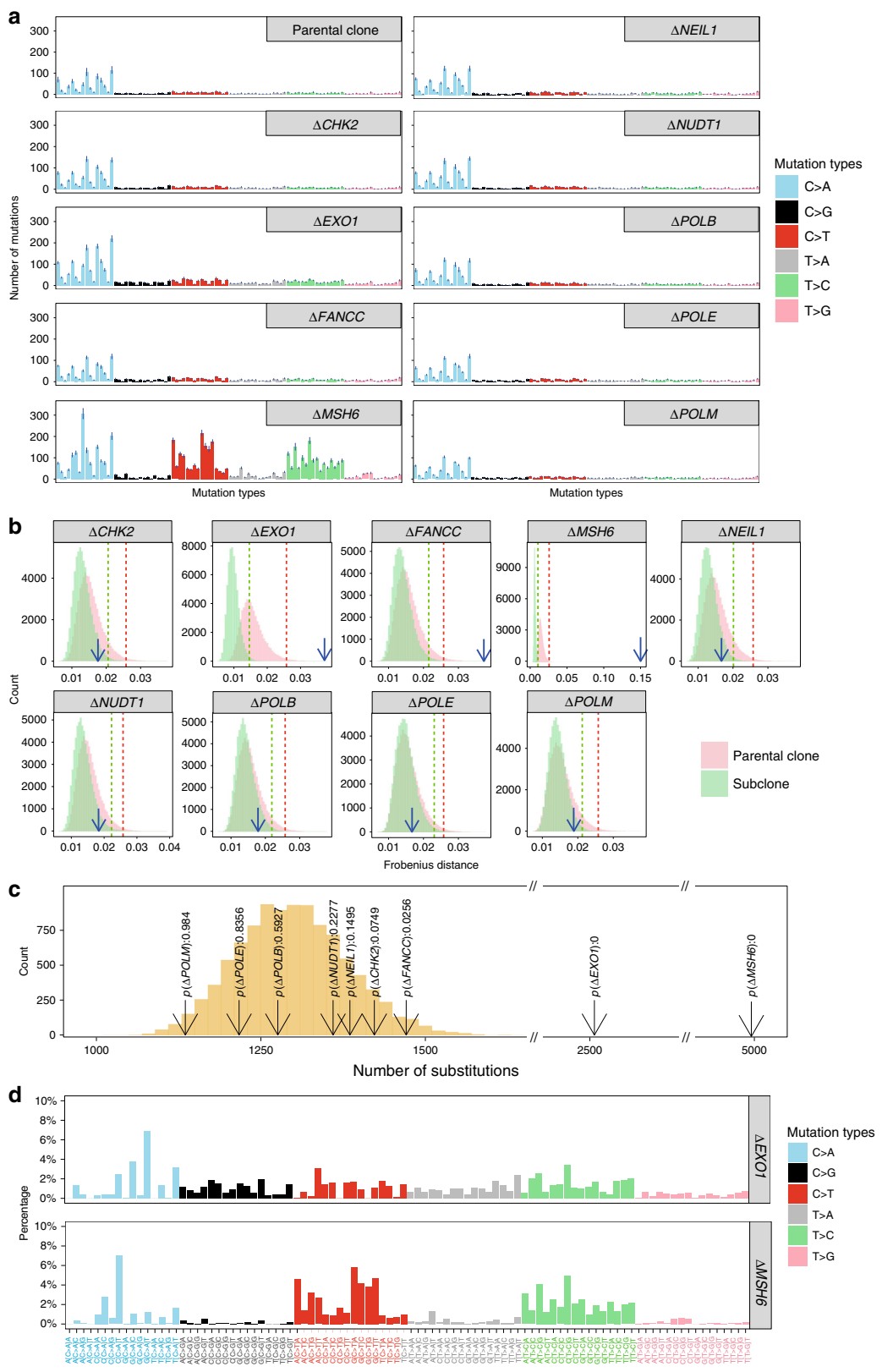

creates distinctive mutation patterns, or mutational signatures, conclusively validating the abstract concept of mutational signatures in human cancers. Furthermore, single gene targeting in vitro in some cases generated not just one but multiple mutational signatures, buttressing previous reports that multiple in vivo cancer-derived signatures could arise from single gene defects such as in *BRCA1/BRCA2*[17]. This is likely to be due to the multitude of compensatory DSB repair pathways that are brought into play in the absence of conservative, error-free HR and due to some activity of translesion synthesis. Whatever are the

mechanisms that underpin these observations, this is important authentication—because multiple mutational signatures are now starting to be exploited as a principle for designing clinical biomarker assays[17]. This notion of using multiple signals as a biomarker would predict more sensitive and more specific tumour stratification—critical for clinical trials that are currently still largely based on single-channel assays with all their attendant limitations.

**Similarities between experimental and cancer signatures**. When mutational signatures were first mathematically extracted from cancers, several mutational signatures were found to be associated with inactivation of DNA repair genes. To investigate how in vitro experimentally-generated mutational signatures of gene knockouts compared with in in vivo cancer-derived signatures, we calculated cosine similarities between the in vivo and in vitro mutational signatures for substitutions (Fig. 6a) and rearrangements (Fig. 6b) (cancer-derived indel signatures are not available). Then, we compared overall mutational profiles of knockouts with those of patient cancers.

The substitution signatures of MSH6 and EXO1 knockouts were compared with cancer-derived 30 COSMIC signatures (http://cancer.sanger.ac.uk/cosmic/signatures). The MSH6 knockout signature is most similar to COSMIC signature 20 with cosine similarity of 0.91 (Fig. 6a), although there are relatively high cosine similarities when compared to other cancer-derived signatures associated with MMR-deficiency (all ⩾0.6). The EXO1 knockout substitution signature is most similar to COSMIC Signatures 3 (0.71) and 5 (0.71). Whole genome profiles of experimentally-generated gene knockouts bear uncanny resemblances to whole genome profiles of primarily repair-deficient tumours (Fig. 6c). The MSH6 knockout, for example, bears striking similarity to those in MMR-deficient tumours—characterised by C>T and T<C substitution signatures and high burden of indels at polynucleotide repeat tracts (Supplementary Fig. 5). By contrast, the FANCC and EXO1 knockouts are more similar to HR-deficient cancers; defined by general genomic instability and an excess of deletions with microhomology at the breakpoint junction (Fig. 6c, Supplementary Figs. 6 and 7). This is an interesting observation because although both of these proteins are not typical HR genes, they do play a role in promoting HR repair of DNA double-strand breaks. These data also provide additional experimental evidence to support how cancers that are deemed to be "HR-deficient", can be sub-classified further genetically.

In a previous analytical exercise exploring structural variation in breast cancer, six classes of rearrangement signatures were identified[15], including two types of tandem duplication signatures—Rearrangement Signature 3 (RS3) comprising short (<10 Kb) tandem duplications and enriched in BRCA1-null tumours and Rearrangement Signature 1 (RS1) comprising long (>100 Kb) tandem duplications, not associated with BRCA1 mutations although a genetic cause has not been identified. The rearrangement signatures of EXO1 and FANCC knockouts were compared with cancer-derived rearrangement signatures (RS1-RS6). The EXO1-knockout rearrangement signature is strikingly similar (0.93) to RS1 which is defined by long tandem duplications (Fig. 6b). By contrast, the FANCC-knockout rearrangement signature shows little similarity (0.09) to RS1, and instead shows elements of RS3 (0.43) and RS5 (0.59), which have short tandem duplications and deletions. Hence, we show that these rearrangement signatures are not just mathematical abstractions but indeed separate biological entities—that is, the two tandem duplication patterns, namely RS1 and RS3, are able to be recreated by knocking out disparate genes. The FANCC knockout rearrangement pattern comprised mainly short tandem duplications and short deletions (<10 Kb) and also had other rearrangement classes but essentially echoed those of BRCA1-null cancers (Fig. 6c and Supplementary Fig. 6). This is consistent with the role played by BRCA1 in HR, downstream of the FA pathway[46,58]. By contrast, the EXO1 knockout rearrangement signature was dominated by medium-to-long tandem duplications emulating the alternative cohort of genomically unstable (but BRCA1-intact) tumours (Fig. 6c and Supplementary Fig. 7).

**Genomic architecture of experimentally-generated signatures**. Previous analyses of breast-cancer-derived mutational signatures revealed diverse relationships with replicative strand and replicative time domains, as well as transcriptional strands. We thus explored whether experimentally-generated mutational signatures mirrored are thereby validated these mathematically-derived observations.

Of the experimentally-generated mutational signatures, first, we did not find evidence of transcriptional strand bias (Fig. 7a and Supplementary Fig. 8). Second, replication strand asymmetry was not observed for the signatures caused by knockouts of EXO1, though it was observed for the C>T/G>A (1.27 fold, p value = 0.021, t test) and T>C/A>G (1.38, p value = 0.018, t test) components of the MSH6 knockout (Fig. 7b). This interesting bias was consistent with the observation that MMR deficiency associated mutational signatures 6, 20 and 26 have either an excess of damage to G and T on the lagging replicative strand or C and A on the leading replicative strand (Fig. 7c). This implied that MSH6 must have a particular role in directing the repair of damage of these nucleotides during replication. Third, while EXO1 knockout mutational signatures were consistently increased in regions of the genome associated with late replication, the mutational signature of MSH6 demonstrated a

**Fig. 3** Determination of substitution mutational signatures in gene knockouts. **a** Profile of 96 mutation types (6 types of substitution ∗ 4 types of 5' base ∗ 4 types of 3' base) of parental clones and DNA repair gene knockouts. A strong background signature is observed in all samples. The substitution spectrum of each sample is shown in Supplementary Fig. 2. Error bars were referred to as standard error of means ($n = 7$). **b** Discrimination of mutation spectrum of parental clone and subclones. Bootstrap sampling method was used to construct a population of parental clones. The distribution of distance of parental clone replicates to the centroid of parental clones is shown as the pink histogram. The red dashed line indicates a cutoff ($d_{pc\_0.01}$) where 99% replicates are within this distance to the centroid of parental clones. The distribution of subclone replicates is shown as the light green histogram. The green dashed line indicates a cutoff ($d_{sc\_0.01}$) where 99% subclones are within this distance to the centroid of all subclones. The blue arrow indicates the distance ($d_{ps}$) between centroid of subclones to the centroid of parental clones. A knockout is considered to have an effect on the substitution spectrum, when $d_{ps} > d_{pc\_0.01}$ and $d_{ps} > d_{sc\_0.01}$ are observed, e.g., EXO1, FANCC, MSH6. **c** Identification of mutation number increase in subclones due to gene knockout. From (**b**), one can discriminate the knockouts that do not generate mutational signatures. The number of mutations in these knockout backgrounds can be used as a baseline; through bootstrap sampling method, we obtained the distribution of the number of mutations in subclones in a wildtype background and, therefore calculated the p value of mutation number of each knockout. EXO1 and MSH6 show significantly elevated mutation numbers as well as mutational profile change. **d** Substitution signatures of EXO1 and MSH6 knockouts. The mutational signatures associated with gene knockouts are obtained by removing the substitution profile of parental clones from the mean of the substitution spectrum of the seven subclones

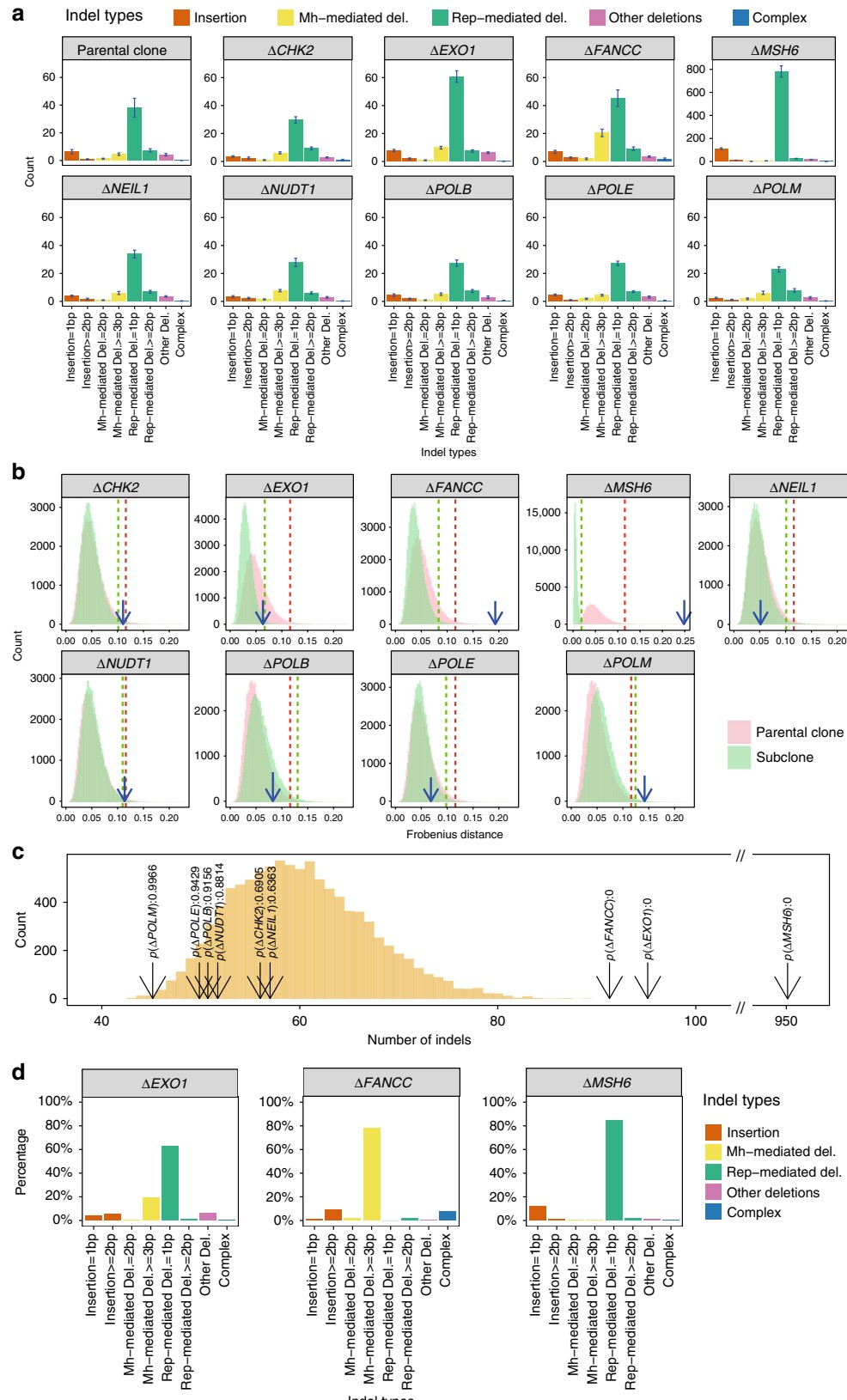

notably flatter slope, with more mutations early in replication compared to the other knockout signatures (Fig. 7d). This strikingly echoed in vivo observations—a base substitution signature associated with tumour MMR deficiency also exhibits a flattened profile across replication timing domains, unlike most other substitution signatures in breast cancer[13]. Crucially, this result from an experimentally-generated knockout of *MSH6* provided support for a previous hypothesis that MMR activity is essential for reducing mutagenesis in gene-rich, early replicative domains. When abolished, the protective role usually played by

MMR on mutagenesis in these regions, is lost, thus resulting in the excess of mutations in early domains and a flattened replication timing profile[13]. In conclusion, our findings collectively show that mutational signature behaviours across genomic architecture are corroborated by in vitro studies.

## Discussion

The gene-edited human cell-based model system used here has permitted validation of the mutational signatures concept across all classes of mutations. This system, however, is not without issues. A challenge posed by the considerable cell culture-related signature resulted in an encumbered signal-to-noise ratio. Here, we combine the experimental set-up with algorithmic developments in order to successfully view mutational patterns generated by knockout of DNA repair genes. These principles lend themselves to a thorough, systematic screen of all genes involved in maintaining genome integrity and of all potential genotoxic agents in order to comprehensively understand the repertoire of mutational signatures in human cells.

We found that in our experimental setup, not all knockouts of genes associated with DNA repair produced detectable mutational signatures. While this could reflect lack of a mutational signature, it is also possible that some gene knockouts produce signals that are too weak to be detected under these experimental conditions. They require intensification through elevating mutation rates. One way this could be achieved is by increasing cumulative time in culture—but the data here already suggest that mutation accumulation rates are variable between genes and a one-size-fits-all approach will therefore always have its limitations. Alternatively, increasing DNA damage experimentally (using acute or chronic regimes) could help to amplify mutagenesis. However, mutational signatures spawned through assisted methods have arisen under subtly different conditions and should be interpreted with this in mind. Using alternative isogenic models that are more permissive for mutagenesis (e.g., MEFs) could also help to increase mutation rates. However, using different cell-based systems with different genetic backgrounds could result in diverse mutational signatures, if similar studies are performed. Lastly, because of the nature of growing cells in culture, it is possible that this is associated with some loss of insights. Copy number changes are often poorly tolerated in cell-based systems and copy number patterns may perhaps be under-represented using these approaches.

Nevertheless, we present a proof-of-principle, demonstrating how experimentally-generated mutation patterns recapitulate those seen through analysis of primary tumours, thus authenticating the abstract concept of mutational signatures. Our findings also validate previously observed mutational signature relationships with replication, both spatially and temporally. We also note that our findings have also highlighted how a single gene defect is not restricted to creating one mutational signature—it can engender multiple mutational signatures of different classes.

The converse is also true: a mutational signature may not necessarily reflect a defect in a single gene, as it could arise through dysregulation of a number of related genes in a pathway. Herein, we have conclusively demonstrated in vitro that endogenous mutational signatures are a direct, mechanistic read-out of pathway dysfunction and could thus be used as biomarkers of pathway dysregulation even in the absence of knowing the precise gene defect or even which gene is compromised.

## Methods

**Culture conditions**. HAP1 cells were grown in Iscove's Modified Dulbecco's Medium (IMDM; GIBCO), containing L-Glutamine and 25 mM HEPES and supplemented with 10% fetal bovine serum (FBS) and 1% Penicillin/Streptomycin (P/S). Cells were grown at 37 °C, with 20% oxygen and 5% carbon dioxide. HAP1 cells were passaged every 3 days and maintained sub-confluent for 1 month. The cell lines were tested negative for mycoplasma contamination using MycoAlert Mycoplasma Detection Kit. HAP1 is not listed in the database of commonly misidentified cell lines by ICLAC. The parental HAP1 cell line has been characterized and authenticated by our collaborators at Horizon Genomics.

**Gene editing by CRISPR-Cas9**. CRISPR-Cas9 knockouts were generated in collaboration with Horizon Genomics. HAP1 cells were transfected with a Cas9 expressing plasmid, a guide RNA (gRNA) expressing plasmid and a plasmid conferring Blasticidin resistance, using Xfect (Clontech). Guide RNA sequences were 5′-AGGTAAAGCTGGCTTTCGAG-3′ (CHK2), 5′-ATCCATCAAATACG AGAAT-3′ (EXO1), 5′-GCCAACAGTTGACCAATTGT-3′ (FANCC), 5′-CCAAG ATGGAGGGTTACCCC-3′ (MSH6), 5′-TGCCCACCTGCGCTTTTACA-3′ (NEIL1), 5′-TTCGGGGCCGGCCGGTGGAA-3′ (NUDT1), 5′- GAGCAAACGGA AGGCGCCGC-3′ (POLB), 5′-AGTTTCGGCACTCAAGCGCC-3′ (POLE), and 5′-ACAGGCCTGGCGCGCTCCAA-3′ (POLM).

Subsequently, the cells were treated with 20 μg/ml Blasticidin for 24 h in order to eliminate untransfected cells. After 5–7 days of recovery from Blasticidin selection, clonal cell lines were isolated by limiting dilutions.

**Sanger sequencing**. Genomic DNA was extracted using Viagen Bitoech DirectPRC Lysis Reagent (Cell) adhering to the protocol provided by the manufacturer. The genomic region targeted by the gRNA was amplified using the primers and PCR amplification conditions provided below. Frameshift mutations were identified using Nucleotide BLAST against the reference genome GCF_000001405.33. Clones with frameshift mutations were selected as parental cell lines.

Forward primers (For) were 5′-TCAAAGATGCCCCAAAATTTTCCAT-3′ (CHK2-For), 5′-CTCGTAAGTATCCAAGGCAGGATTT-3′ (EXO1-For), 5′-CA AACCTACACACACACATACATGGAC-3′ (FANCC-For), 5′-TGGCAGTAGTGAC TCTTACCTGTAT-3′ (MSH6-For), 5′-TGGCCAGCCAGTTTGTGAAT-3′ (NEIL1-For), 5′-GCTGGGGAGTTACAGCATACC-3′ (NUDT1-For), 5′-ACTTG TGAATAATTTTGTGTGGGTCA-3′ (POLB-For), 5′-CACTCTTTAGATAA GGACCACGCTA-3′ (POLE-For) and 5′-TCGCCCTAATTAATAGCACCCTT TA-3′ (POLM-For).

Reverse primers (Rev) were 5′-CTTTGTTTTTCCCTCTAGTGGTGC -3′ (CHK2-Rev), 5′-ATCATAGGGTACTAAGGTGCTGAAC-3′ (EXO1-Rev), 5′-ACTAAACAAGAAGCATTCACGTTCC-3′ (FANCC-Rev), 5′-AATGCCA GAAGACTTGGAATTGTTT-3′ (MSH6-Rev), 5′-TGGTACTCCTGCAAGA CACA-3′ (NEIL1-Rev), 5′-GAAACCAAGGGTGTGGCCCTA-3′ (NUDT1-Rev), 5′-CAGATCATAAGCTATGGAAGGGTGA-3′ (POLB-Rev), 5′-AGAGCAAGA CTCCGTCTCAAAAA-3′ (POLE-Rev) and 5′-CGGAGTTTCCCTCTGCGTT-3′ (POLM-Rev).

PCR amplification: heat lid to 110 °C; start reaction with 94 °C for 2 min; loop 35 × (94 °C for 30 s; 55 °C for 30 s; 68 °C for 1 min), then finish with 68 °C for 7 min.

**Fig. 4** Determination of indel signatures in gene knockouts. **a** Indel spectra of parental clones and DNA repair gene knockouts are represented by a 8-channel indel profile which takes the type, length of indel motif and the characteristics at the indel junction into account: 1 bp insertion, >=2 bp insertion, 2 bp microhomology-mediated deletion, >=3 bp microhomology-mediated deletion, 1 bp repeat-mediated deletion, >=2 bp repeat-mediated deletion, other deletions and complex indels. Error bars were referred to as standard error of means ($n = 7$). The indel spectrum of each sample is shown in Supplementary Fig. 3. **b** Distribution of bootstrapped indel spectra of parental clone (pink) and subclones (green). FANCC, MSH6 and POLM show significant changes in indel spectrums. **c** Comparison of indels numbers among subclones. The cyber yellow distribution is generated by bootstrapping the indel number of knockout subclones without significant changes in indel profiles. EXO1, FANCC and MSH6 show significant increases in indel numbers, indicating the effect of gene knockout on indels. In contrast, although POLM shows a detectable indel spectrum shift, it did not show a clear increase in indel number ($p$ value = 0.9966). Hence, POLM cannot be determined to have an indel signature. **d** Indel signature of EXO1, FANCC and MSH6. Indel signature of EXO1 is similar to the culture indels signature. Indel signature of FANCC is dominated by microhomology-mediated deletions of 3 bp or more. Indel signature of MSH6 is dominated by 1 bp deletions at poly-nucleotide repeat tracts

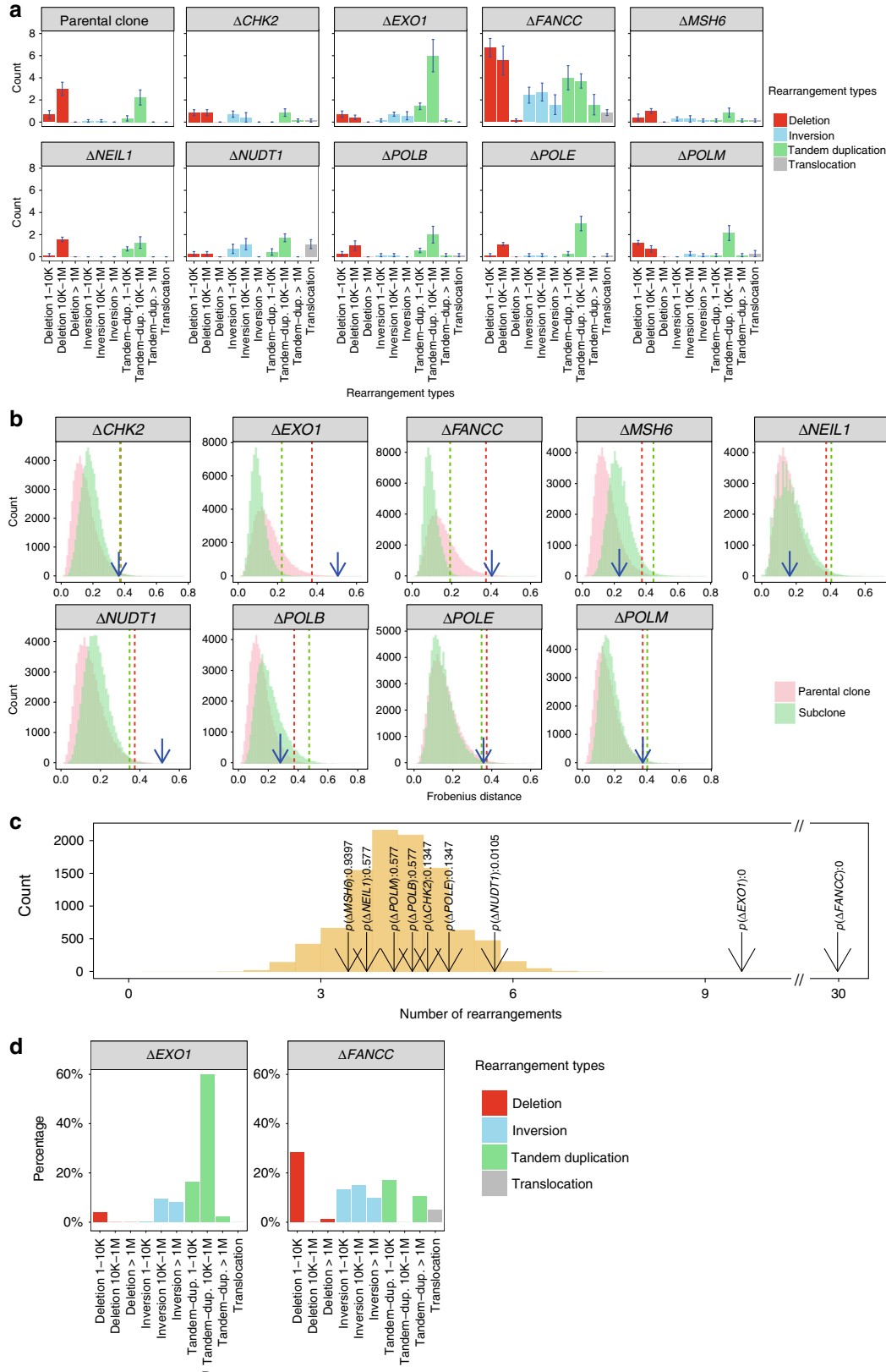

**Proliferation assay**. Knockout cell lines were plated in triplicates at a density of $0.32 \times 10^6$ cells ml$^{-1}$ and allowed to proliferate. Every second day, cells were dissociated with Trypsin-EDTA (Gibco), living cells were counted using CASY Cell Counter and Analyzer system (Innovatis), and replated at 1:2, 1:3 or 1:4 dilutions, depending on the growth rate of the cell line. The experiment was carried out for 10 days. Proliferation was plotted for each time point considering the dilution rates. The average growth rate is a mean over 10 days.

**Protein extracts and immunoblotting**. Cell extracts were prepared using RIPA lysis buffer (NEB) with protease (Sigma) and phosphatase (Sigma) inhibitors. Immunoblots were performed using standard procedures. Samples containing proteins were separated using SDS PAGE 4–12% gradient gels (Invitrogen) and transferred onto nitrocellulose membranes. The membranes were incubated with primary and secondary antibodies. The primary antibodies were NUDT1 (NB100-109, Novus Biologicals), CHK2 (05–649, Millipore), POLM (C1, Santa Cruz),

EXO1 (A302-639A, Bethyl Laboratories), FANCC (MABC524- clone 8F3, Millipore), POLE (GTX132100, GeneTex), Actin (A5060, SIGMA), NEIL1 (12145-1-AP, Proteintech), POLB (ab26343, Abcam), and MSH6 (D60G2, Cell Signalling). Catalogue numbers and working dilutions for antibodies are provided in Supplementary Table 1. Uncropped immunoblot images are shown in Supplementary Fig. 9.

**DNA library preparation and sequencing.** Five hundred nanogram of genomic DNA was fragmented (average size distribution ~500 bp, LE220, Covaris Inc), purified, libraries prepared (Agilent SureSelect XT custom kits, Agilent Technologies), and index tags applied (Sanger 168 tag set). Index tagged samples were amplified (6 cycles of PCR, KAPA HiFi kit, KAPA Biosystems), quantified (dsDNA BR assay, HS assay, *Thermo Fisher Scientific*), normalized (~0.85 ng/μl), then pooled together in an equivolume fashion. Pooled samples were submitted to cluster formation for HiSeq ×10 sequencing (32 lanes, 150 bp PE read length, Illumina Inc). The average sequencing coverage is 15-fold for all samples given that HAP1 is a haploid cell line. The details of sequence coverage for all clones and subclones are provided in Supplementary Data 5.

**Alignment and somatic variant-calling.** Short reads were aligned to human reference genome GRCh37/hg19. Somatic substitutions, indels and rearrangements in clones and subclones were called by CaVEMan[59] (http://cancerit.github.io/CaVEMan/), Pindel[60,61] (http://cancerit.github.io/cgpPindel) and BRASS[15] (https://github.com/cancerit/BRASS), respectively.

De novo somatic mutations of substitutions, indels and rearrangements in subclones were obtained by removing all mutations seen in parental clones. The summary of de novo somatic mutations for each gene knockout is provided in Supplementary Data 6.

**Determination of mutational signatures for gene knockouts.** The mutational landscape of a cell over a certain period of time reflects a balance point between DNA damage and repair processes in the cell. Exposure to exogenous mutagenic agents or abrogation of DNA repair activity could affect this balance, thereby inducing changes in the mutational landscape. Based on this principle, if the knockout of a gene effectively generates a mutation pattern, then one could observe two changes: First, a shift in the mutational spectrum of cells between subclones and parental clones (shown schematically in Fig. 2a); Second, a change in numbers of mutations in subclones when compared to background (Fig. 2b).

To conclusively identify an effect of a gene knockout, three steps are required: (1) Detecting a qualitative difference between mutational spectra of knockout subclones and that of parental clones; (2) Detecting a quantitative difference in numbers of mutations. (3) Extracting knockout signature. Figure 2c demonstrates the workflow. A more detailed method is described below.

In step 1, we applied a bootstrap resampling method on parental clones and subclones, and calculated the Frobenius distance between parental clones and subclones to quantify the difference between the mutational spectrum of parental clone (without gene knockout effects) and that of subclones (with gene knockout effects).

First, mutation profiles for parental clones ($M_p$) and subclones ($M_s$) for each gene KO were defined as:

$$M_p = \begin{bmatrix} m_p^1 \\ \vdots \\ m_p^K \end{bmatrix} \text{ and } M_s = \begin{bmatrix} m_{s1}^1 & \cdots & m_{s7}^1 \\ \vdots & \ddots & \vdots \\ m_{s1}^K & \cdots & m_{s7}^K \end{bmatrix},$$

where $m$ is the mutation number of each mutation feature in each sample, $p$ and $s$ refer to the parental and subclones of different gene knockouts respectively.

The substitution spectrum is made up of a 96-channel vector ($K = 96$), where for each of the six classes of C>A, C>G, C>T, T>A, T>C and T>G, the flanking 5′ and 3′ sequence context for each of the mutated bases is also taken into account (6 types of substitution ∗ 4 types of 5′ base ∗ 4 types of 3′ base = 96 channels). For indels, the profiles are made up of eight features ($K = 8$), including 1 bp insertion, >= 2 bp insertion, 2 bp microhomology-mediated deletion, >= 3 bp

microhomology-mediated deletion, 1 bp repeat-mediated deletion, >= 2 bp repeat-mediated deletion, other type of deletion and complex indels, are used. For rearrangements, ten mutation features ($K = 10$) are employed: 1–10 Kb, 10 Kb–1 Mb, and >1 Mb sized deletions, inversions and tandem-duplications respectively and translocations. The profile of substitutions, indels and rearrangements for all samples are shown in Supplementary Figs. 2–4, respectively.

Second, a bootstrap distribution for parental clones was generated. Bootstrap resampling was applied to each parental clone to generate 7000 replicates where the frequency of each mutation type corresponded to its probability in the clone multiplied by the total counts. In total, for nine parental clones, 63,000 replicates are generated. From 63,000 replicates, seven samples are randomly selected and the normalized distance between the centroid of the seven chosen replicates and the centroid of original parental clones, is calculated as $d_{pc}$. By repeating this step 10,000 times, we obtain a distribution of $d_{pc}$ (shown in Fig. 2d), and the distance associated with $p$ value = 0.01, $d_{pc\_0.01}$, is identified.

Third, bootstrap distributions for subclones of knockouts were generated. The application of bootstrapping on subclones is similar to that of parental clones, see Fig. 2e. For each knockout, 63,000 replicates of subclones are generated (9000 replicates * 7 subclones). Nine replicates are randomly chosen from 63,000 replicates and are used to calculate the normalized distance between the centroid of replicates and the centroid of original subclones, $d_{sc}$. The distribution of $d_{sc}$ is therefore obtained by repeating the previous step for 10,000 times and the threshold distance with $p$ value = 0.01, $d_{sc\_0.01}$, can be calculated.

Finally, changes in mutational spectrum between parental clones and subclones were determined. For each of the gene knockouts, the distance between centroid of parental clones and centroid of subclones ($d_{ps}$) is compared with $d_{pc\_0.01}$ and $d_{sc\_0.01}$. The criterion to determine whether the mutational spectrum associated with a given gene knockout is significantly different to the parental clone is $d_{ps} > d_{pc\_0.01}$ and $d_{ps} > d_{sc\_0.01}$, see Fig. 2a.

Step 2 involves determination of increase of mutation number associated with a gene knockout. Aggregated mutation numbers of gene knockouts that do not have a change in mutation spectrum (results from step 1) are used to construct a distribution of baseline mutation counts (i.e., no effect of gene knockout), as shown in Fig. 2f. According to this distribution, a $p$ value of aggregated mutation number of each gene knockout can be calculated. Gene knockouts with $p$ value < 0.01 are considered to have a significantly elevated mutation count, indicative of mutational signatures associated with abrogation of these genes.

In step 3, we extracted knockout signatures based on quantile analysis. The mutational spectrum of subclones can be seen as a linear combination of the mutational spectrum present in parental clones (background mutagenesis) and the mutational spectrum associated with the specific gene knockout:

$$\overline{M_s} \approx e_p \times \overline{P_p} + e_{ko} \times P_{ko}$$

where $\overline{P_p} = \sum_p M_p / \sum_p \sum_k m_p^k$ and $\overline{M_s}$ is the centroid of seven subclones of each knockout gene. ko refers to different gene knockouts. $e_p$ and $e_{ko}$ are the number of mutations caused by parental clone signature and knockout gene signature respectively.

Hence, once a knockout gene is considered to have a mutational signature, its signature ($P_{ko}$) can be obtained by removing mutations associated with parental clones from the mutation profile of the subclone:

$$P_{ko} \approx (\overline{M_s} - e_p \times \overline{P_p}) / e_{ko}$$

The detailed steps are as described below:

First, we generated bootstrap distributions of subclones. For each knockout gene, 10,000 replicates of subclones are generated to construct a distribution of mutation number in $k^{th}$ of $K$ features of each of the subclones. According to that distribution, the upper and lower boundaries (99% CI) for each $k^{th}$ feature are identified.

Second, the initial status is assumed that there is no knockout signature, i.e., background exposure, $e_p$, is the total mutation number of subclones. Thus, the background signature profile, $e_p \times \overline{P_p}$, can be calculated. Each number in $k^{th}$ of $K$ features of background signature profile was compared with the upper and lower boundaries of each $k^{th}$ feature of subclones calculated from step 1. For each step,

**Fig. 5** Determination of rearrangement signatures in gene knockouts. **a** The rearrangement spectra of parental clones and DNA repair gene knockouts are represented by a 10-channel profile that takes the type and length of rearrangements into account. The rearrangement spectrum of each sample is shown in Supplementary Fig. 4. Error bars were referred to as standard error of means (n = 7). **b** Distribution of bootstrapped rearrangement spectra of parental clone (pink) and subclones (green) of the knockouts. *EXO1*, *FANCC* and *NUDT1* knockouts show significant changes in their rearrangement profiles. **c** Identification of elevated rearrangement numbers in knockouts. *EXO1* and *FANCC* knockouts show high number of rearrangements (p value <= 0.01), while *NUDT1* has a p value of 0.0105, which is at the border of our threshold. To be conservative, *NUDT1* is not determined to have a rearrangement signature. **d** Rearrangement signature of *EXO1* and *FANCC*. The rearrangement signature associated with knockout of *EXO1* is characterised by median tandem duplications (10 kb–1 Mb). The rearrangement signature associated with knockout of *FANCC* is characterised by short deletions (1–10 kb), deletions and tandem duplications of 1–10 kb and 10 kb–1 Mb

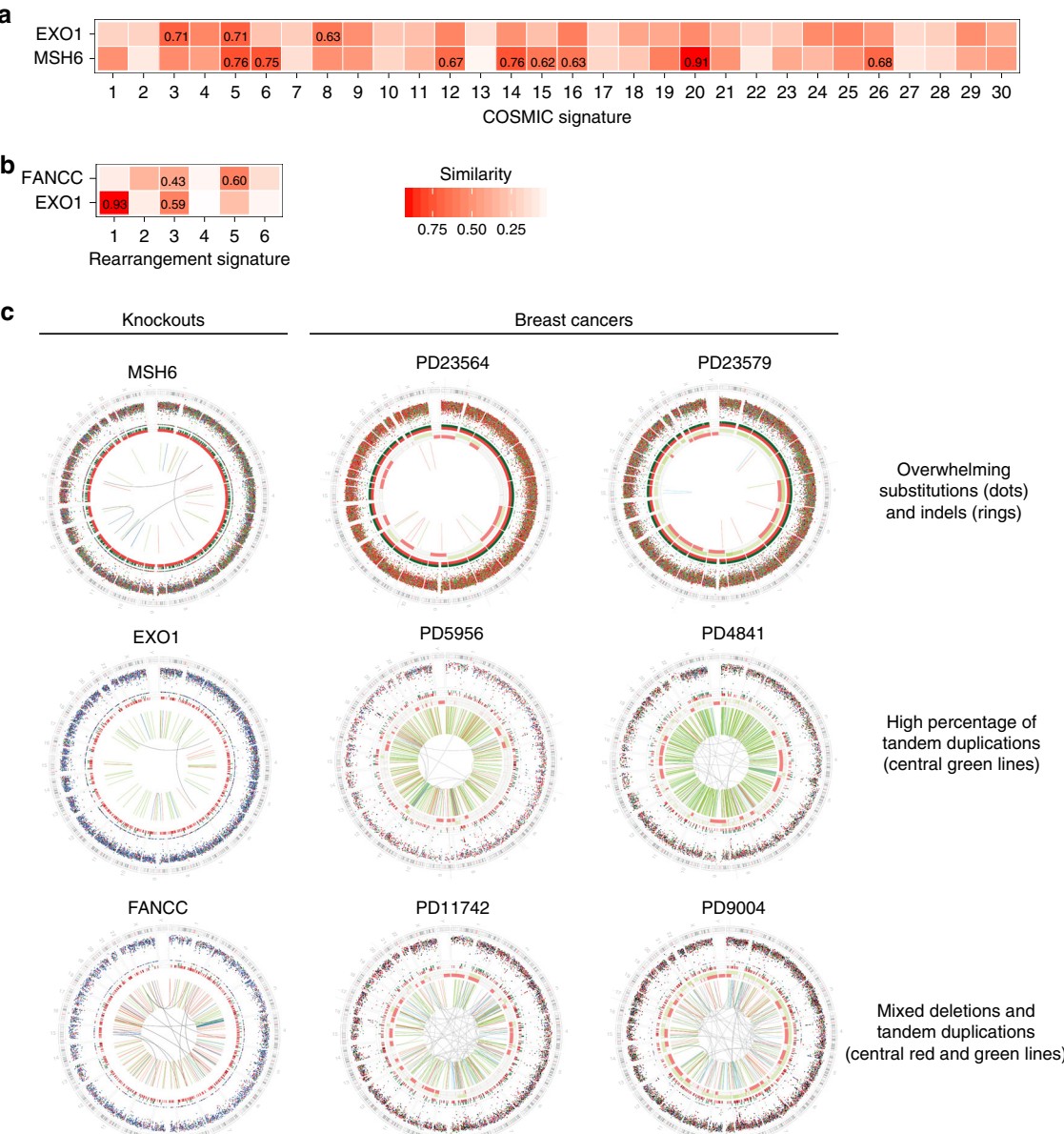

**Fig. 6** Comparison of mutational signatures between cancer (in vivo) and knockouts (in vitro). **a** Cosine similarity between 30 COSMIC substitution signatures (http://cancer.sanger.ac.uk/cosmic/signatures) and *EXO1/MSH6* knockout substitution signatures. **b** Cosine similarity between six cancer-derived rearrangement signatures and *EXO1/FANCC* knockout rearrangement signatures. **c** Genome plots of *MSH6*, *EXO1* and *FANCC* knockouts and of cancer samples. Genome plots show somatic mutations including substitutions (outermost, dots represent six mutation types: C>A, blue; C>G, black; C>T, red; T>A, grey; T>C, green; T>G, pink), indels (the second outer circle, colour bars represent five types of indels: complex, grey; insertion, green; deletion other, red; repeat-mediated deletion, light red; microhomology-mediated deletion, dark red) and rearrangements (innermost, lines representing different types of rearrangements: tandem duplications, green; deletions, orange; inversions, blue; translocations, grey). Genome plot of *MSH6/EXO1/FANCC* HAP1 knockouts are aggregations of seven subclones. PD23564 and PD23579 are breast cancers with microsatellite instability which is resulted from impaired mismatch repair. PD5956 and PD4841 are two breast cancers that would historically have been termed as having HR deficiency but are enriched for rearrangement signature 1 and distinct from *BRCA1/BRCA2*-mutated cancers. PD11742 and PD9004 are two breast cancers with *BRCA1/BRCA2*-null HR deficiency

100 bootstrapping background exposure profiles are generated, and if there are at least five parental signature profiles fall within the boundary of subclones, the current background exposure is determined as the final background exposure, and iteration stops. Otherwise, $e_p$ will reduce by 1 in the next step and the newly constructed status will be compared with mutational profiles of subclones.

Third, once the background exposure, $e_p$, is identified from step 2, the exposure associated with a knockout is thus obtained by subtracting parental exposure from centroid of subclones.

**Topography of mutations associated with knockout genes**. We explored the relationships between genomic features, e.g., DNA replication and transcription, and mutations associated with knockout genes. Reference information of replicative strands and replication timing regions were obtained from the ENCODE

project Repli-seq data (https://www.encodeproject.org/)[62]. Regions of protein coding gene in the genome were used to assign transcriptional strand coordinates. Here, all substitutions are represented in pyrimidine context and the coordinates of transcriptional and replicative strands are given on the +strand of the reference genome, therefore the transcriptional/replicative strand information associated with each substitution is adjusted to the pyrimidine-based mutation, e.g., a G>C mutation on the transcribed strand is described as a C>G mutation on the non-transcribed strand.

**Code availability**. The code for determination and extraction of knockout signatures associated with this study is available from corresponding author (S.N.-Z.) upon request.

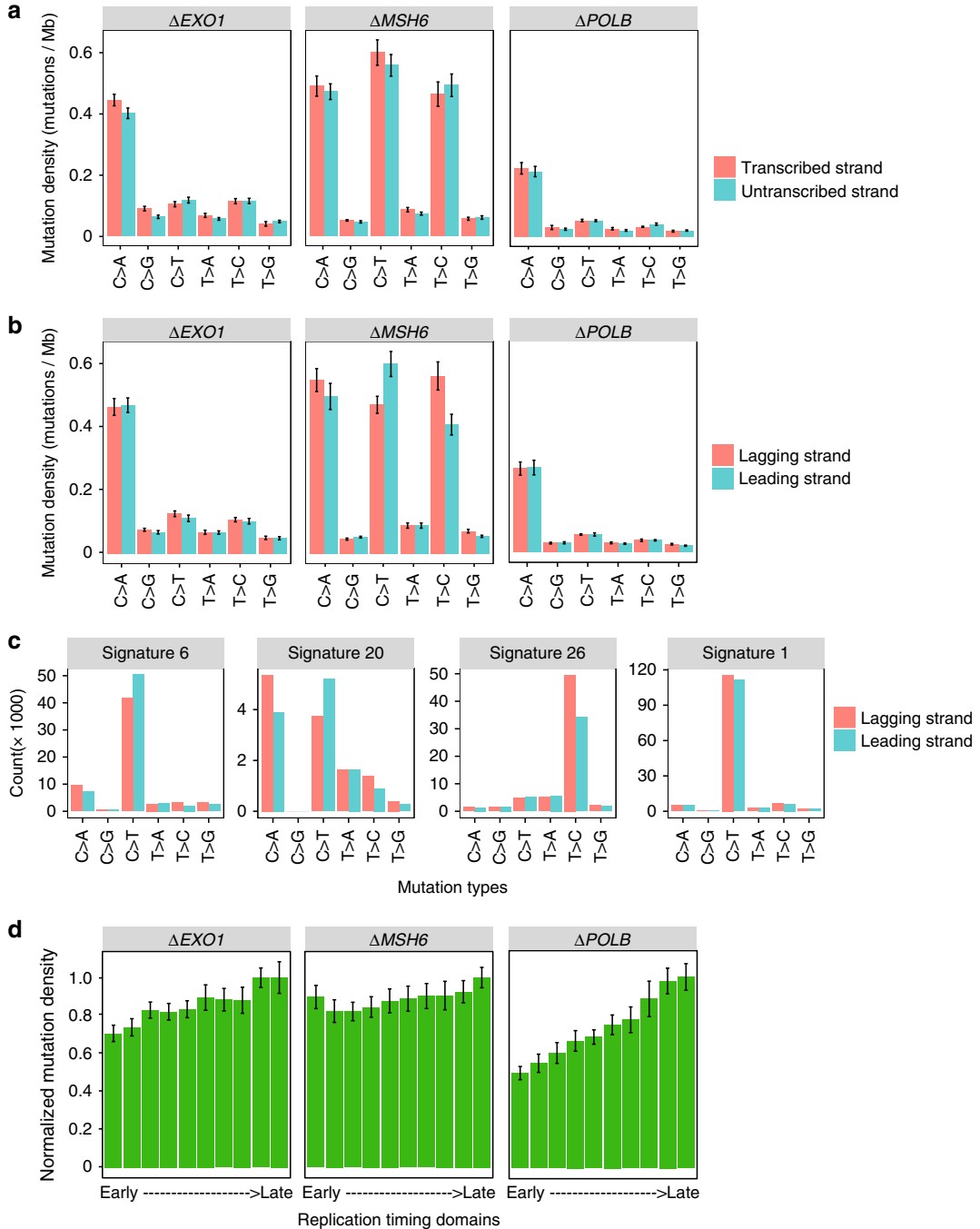

**Fig. 7** The topography of experimentally-generated mutations of *EXO1*, *MSH6* and *POLB* knockouts. *POLB* does not show a mutational signature in substitutions. It is shown here as a contrast against *EXO1* and *MSH6* signatures. The topography of mutational signatures associated with the remaining six knockout genes is shown in Supplementary Fig. 8. **a** Histograms exploring transcriptional strand asymmetry. **b** Histograms exploring replication strand asymmetry. **c** Histograms showing replicative strand asymmetry of mutational signatures in breast cancers. Twelve mutational signatures were identified from 560 breast cancers[15]. Here only four signatures are shown: Signatures 6, 20 and 26 are associated with mismatch repair (MMR) deficiency; Signature 1 is associated with hydrolytic deamination of methylated CpG is shown as a contrast. **d** Distribution of normalized mutation density across the replication timing domains. The G2/S phase was separated into ten replication timing domains[13]. Mutation densities in replication timing domains were corrected for genomic size of each domain

**Data availability**. All mutation data can be obtained from: ftp://ftp.sanger.ac.uk/pub/cancer/Zou_et_al_2017

All other remaining data are available within the Article and Supplementary Files, or available from the authors upon request.

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

## Acknowledgements

**Funding:** Wet-lab consumables were provided by the J.I.L. lab, that is funded by the Austrian Academy of Sciences and was additionally funded by a Marie-Curie Career Integration Grant at the start of the project (Project number: 321602-NonCanATM). Sequencing experiments were funded by a Wellcome Trust Intermediate Clinical Fellowship (WT100183MA).
**Personal funding:** X.Z. is funded by a Wellcome Trust Strategic Award (WT101126/B/13/Z). M.O. is supported by a project grant from the FWF to J.I.L. (Project number P 29763-B27). S.N.-Z. was funded by a Wellcome Trust Intermediate Clinical Fellowship

(WT100183MA) at the start of the project and is now funded by a CRUK Advanced Clinician Scientist Award (C60100/A23916). S.P.J. laboratory is funded by Cancer Research UK (programme grant C6/A18796) and a Wellcome Trust Investigator Award (206388/Z/17/Z). Institute core infrastructure funding is provided by Cancer Research UK (C6946/A24843) and the Wellcome Trust (WT203144).
Project Coordination assistance was received from Dr Rebecca Harris. Mr Marc Wiedner and Dr Jana Stranka assisted with cell culture.

## Author contributions

S.N.-Z., J.I.L. and S.P.J. designed the project. M.O. designed, optimized, coordinated and performed all wet-lab experiments. X.Z. performed all steps of data curation, post-hoc processing and performed downstream bioinformatic analysis. R.H. assisted in coordinating the project. S.N.-Z., X.Z., J.I.L and M.O. wrote the manuscript with comments from all authors.

## Additional information

**Competing interests:** S.N.-Z. has patents filed with the UK IPO and is a scientific consultant for Artios Pharma Ltd. J.I.L. has filed a patent with the European Patent Office (EPO). All other authors declare no competing interests.

