## [Peer Review File · Nature Communications]

Reviewers' comments:

Reviewer #1 (Remarks to the Author):

The authors generated DNA repair gene-knockout cells from a near-haploid human cell line HAP1 using CRISPR/Cas9 system. After culturing these cells, the authors analyzed mutational signatures of subclones of these cells. They found that defects of specific DNA repair genes can lead to some mutational signatures.

The studies are well done and informative.

Minor points:

1) If the authors want to compare BRCA1-mutated clinical samples with KO cells, they should generate BRCA1-KO cells (not FANCC-KO cells or EXO1-KO cells). The authors should explain why they did not make BRCA1-knockout instead of FANCC-knockout.

FANCC, BRCA1, and BRCA2 are different genes. Although the proteins encoded by these genes are involved in a common ICL repair pathway (the FA-BRCA pathway), their involvements in HR are different. For example, it is known that FANCC-deficient cells are only mildly deficient in HR. Therefore, we would not expect that the mutational signatures of FANCC-knockout cells be identical to those of BRCA1 or BRCA2-mutated cancer cells. In fact, in Figure S6C, translocation is much more frequently seen in BRCA1-mutated clinical samples than in FANCC-knockout cells. The authors should comment on this.

2) In Figure S1, NEIL1, POLB, POLE blots are not shown. How did the authors confirm the knockouts of these genes?

3) The authors should discuss whether the mutational signatures in these KO HAP1 cells can be generalizable in other cell systems.

Reviewer #2 (Remarks to the Author):

This study presents an in vitro validation of cancer mutational signatures through CRISPR-cas9 mediated knockouts. This is a very interesting and clever paper that addresses a fundamental problem in cancer genomics, namely the gap between the computational analysis of mutational signatures based on genomic data from patient samples, and the underlying biology that gives rise to those signatures. Although the experimental design is well thought and the bioinformatics analysis appropriately carried out, there are a few points that need to be addressed:

1) KO of 6/9 genes (CHK2, NEIL1, NUDT1, POLB, POLE, POLM) did not produce mutational signatures. What do the authors make of this? Is it partly a matter of time, meaning it would take many more cell divisions to detect such signatures? Or is it due to the experimental design, as the authors discuss for POLE for instance? It would be interesting to expand on this.

2) In the case of those genes producing signatures, such as MSH6, the authors here have the opportunity to measure the 'mutation rate' of each signature per generation, however they do not seem to report it. This would be a very interesting measurement to see in the main manuscript.

3) Figure 6 compares mutational signatures experimentally generated by the authors with breast cancer data. However, using circos plots to make this comparison renders the figure unreadable. It is impossible to determine the level of resemblance between the two datasets from this figure. I

appreciate that the in vitro data cannot recapitulate perfectly what happens in humans, but at least a better way of comparing the actual signatures should be presented. Moreover, many more than just a handful of cases should be shown since large datasets of >500 WGS breast cancer samples are available to the authors.

4) There seem to be a protein loading control missing in the western blot in the Supplementary Figure 1. This is essential to distinguish knockout effect from potential protein loading issue.

5) Figure S8 d should be corrected to "c"

Reviewer #3 (Remarks to the Author):

I am not an expert on DNA repair thus I cannot comment on the novelty of the findings or on the computational approaches used.

The data produced here seems very interesting and I would make sure that the raw data is available post-publication.

Regarding the genome editing aspects: The methods used to isolate clones and validate the knockouts are solid. I would be worried about acquiring additional mutations in the clonal expansion steps that would affect mutational signatures as well, e.g. additional mutations in DNA repair genes. Can the authors analyze the data to look for additional loss-of-function mutations in their clones?

In addition, there is no reference to sgRNA off target effects, to minimally address this issue the authors should computationally analyze off-targets of the sgRNAs chosen and show that (1) there is no effect on other genes. (2) there is minimal intergenic off-targets, as a large number of intergenic off-target effects can cause elevated DNA damage that will effect cell viability and maybe also the DNA damage machinery.

To maximally address this issue the authors can show that two sgRNAs that target the same genes (e.g. on two different exons) produce the same result.

Another issue that need to be referenced is the choice of clonal expansion vs. polyclonal experiments. Using lentiviruses to deliver Cas9 and sgRNAs can result in a close to complete KO levels in a polyclonal populations. Would such an approach simplify or complicate the experiments presented in this paper?

Lastly, does any of these gene KOs affect growth rate which would affect the number of cell divisions and the quantitative mutational outcome?

Reviewer #4 (Remarks to the Author):

The authors introduced the concept of mutational signatures several years ago and have produced several landmark papers on this issue since then. In the current paper, they describe proof-of-principle experiments employing CRISPR-Cas9 based gene editing. To this end, they generated targeted CRISPR-Cas9-based knockouts of DNA repair genes in the near-haploid HAP1 cell line to test whether this would result in specific mutational signatures. The design of the experiments was to evaluate mutations that occurred over approximately 45 cellular divisions. From nine gene knockouts (CHK2, EXO1, FANCC, MSH6, NEIL1, NUDT1, POLB, POLE, and POLM) three (EXO1, MSH6 and FANCC) resulted in identifiable mutational signatures (MSH6: substitution and indel signatures; EXO1:

substitution, rearrangement and indel signatures; FANCC: rearrangement and indel signatures). Hence, the major claims of the paper are that the concept of mutational signatures was validated with these experiments, that knockouts of some DNA repair genes may create distinctive mutation patterns, and the demonstration of various mutational signatures caused by a single gene. This is an intriguing manuscript with novel data, which should be of interest to others in the community and the wider field. At the same time there are some issues, which mainly concern comparisons with cancer-derived mutational signatures and which need to be addressed:

Regarding the MSH6 knockout: The authors claim that the mutational signature "bears a resemblance to the multiple substitution signatures that have been associated in cancers, but was not perfectly identical to any of them." Why do they compare the mutational signatures to "cancers" in general; how did it compare to colorectal or endometrium carcinomas, the two tumor entities associated with MSH6 germline mutations? They describe a striking similarity to the MMR deficiency signature in breast cancer. However, the same group has published recently a paper (Cancer Res 77:4755-62), where only 11 of 640 breast cancers were found to be MMR deficient. Apparently, none of them had an inactivation mutation in MSH6; in fact, inactivation mutations in MMR-related genes were found in only six cases. Given the sparse available data on MMR-deficient breast cancers, what data set is the authors' claim of similarity to biological cancers based on?

The authors describe similarities between experimentally generated and cancer-derived mutational signatures as "striking" or "uncanny". However, to this end, Figure 6 is not convincing. Firstly, the scale of the figure is not suited to identify all the differently colored dots. Secondly, there seem to be clear differences between the plots. For example, in Fig. 6d in the outermost circle blue is the dominating color, which is not the case in the corresponding Figs. 6e-f. Similarly, the colors in the second outer circles do not match and neither do the rearrangements in the innermost circles. The same applies also to Figs. 6a-c and Figs. 6g-i. In order to provide convincing evidence for the claim of "striking similarities", either some robust statistics have to be presented or the figure has to be changed. At present, it remains unclear whether the in vitro data have any relevance for the in vivo scenario.

The observation that the same single gene defect in isogenic cell lines under standardized culture conditions can result in different mutational signatures is intriguing and not easy to grasp. Unfortunately, the authors do not discuss this, but they should provide their views how this can be explained.

Response to Reviewers

We thank the reviewers for their comments to improve our manuscript. A point-by-point response to the reviewers' comments is provided below. Statements by the reviewers are given in **bold text** with our response in plain text. Text newly added to the manuscript is shown in **red**.

Reviewer #1

1) If the authors want to compare BRCA1-mutated clinical samples with KO cells, they should generate BRCA1-KO cells (not FANCC-KO cells or EXO1-KO cells). The authors should explain why they did not make BRCA1-knockout instead of FANCC-knockout.

BRCA1 and *BRCA2* are essential DNA repair genes involved in homologous recombination (HR) pathways.

Mammalian cells do not tolerate knocking out of *BRCA1* or *BRCA2* genes. Murine models of *BRCA1/BRCA2* deficiency have shown that they are embryological lethal^{1,2}. The difficulties in generating *BRCA1* or *BRCA2* knockouts (KO) are well-known. Current methods employed to generate *BRCA1/BRCA2* KO include

- Generating hypomorphic alleles instead of full KOs
- siRNA-based / shRNA-based experiments and
- *BRCA1* or *BRCA2* KO on a genetically permissive KO background (such as *TP53* KO).

But these would complicate interpretation in our screen.

In our experiments, we did not set out to use *FANCC*-KO or *EXO1*-KO to replace *BRCA1*-KO at all.

Both *FANCC* and *EXO1* are important genes involved in multiple DNA repair pathways in their own right. They were included in this study because of their distinct repair activities in human cells. In our manuscript, after obtaining our results, we made the comparison between *FANCC*-KO with *BRCA1*-mutated cancers because there are similarities in the signatures that are produced. Indeed, this is probably not a surprise as *FANCC* KO cells have been found to be functionally defective in HR repair in other studies^{3,4}.

However, we realized that this comparison could be confusing to the general reader, thus we have added a sentence on page 8 in the manuscript:

“This is an interesting observation because although both of these proteins are not typical HR genes, they do play a role in promoting HR repair of DNA double-strand breaks.”

2) In Figure S1, NEIL1, POLB, POLE blots are not shown. How did the authors confirm the knockouts of these genes?

We now present immunoblots for all genes within this study (Figure S1b). We additionally confirmed the targeting by Sanger sequencing, as indicated in the legend of Figure 1 and now we include the targeting data in Figure S1a.

We note that using the POLB antibody, we observed a decrease in molecular weight of the protein, even though the Sanger sequencing of the target site indicates a frameshift mutation. This change in molecular weight could imply that there is an alternative translation start site downstream of the successfully targeted Exon 1 within POLB (Figure S1a).

3) The authors should discuss whether the mutational signatures in these KO HAP1 cells can be generalizable in other cell systems.

In the manuscript, we mentioned that *MSH6* knockouts have a near-identical mutational signature in a HAP1 cell line and a human iPS cell line (unpublished data) (second paragraph page 6, also see below response to Reviewer #3 question 3). We believe that for some genes like the ones involved in mismatch repair, the mutational signature produced in different cell lines could be very stable and very similar.

However, it is possible that a KO of a different gene in different cell lines could produce subtly different profiles. In fact, this would be quite an interesting outcome and the likelihood is that there will be variability in results from one gene to another.

In the Discussion, we have suggested that using alternative isogenic models with different genetic backgrounds could result in diverse mutational signatures. To reinforce this further we have modified that section on page 9 slightly to say this:

“Using alternative isogenic models that are more permissive for mutagenesis (e.g., mouse embryonic fibroblasts) could also help to increase mutation rates. **However, using different cell-based systems with different genetic backgrounds could result in diverse mutational signatures, if similar studies are performed.**”

Reviewer #2

1) KO of 6/9 genes (CHK2, NEIL1, NUDT1, POLB, POLE, POLM) did not produce mutational signatures. What do the authors make of this? Is it partly a matter of time, meaning it would take many more cell divisions to detect such signatures? Or is it due to the experimental design, as the authors discuss for POLE for instance? It would be interesting to expand on this.

Indeed, under these experimental conditions, we captured mutational signatures for three out of nine genes. In order to compare and describe data from these experiments, we need to control for some factors. Length of time in culture is one of those factors. By controlling the length of time in culture here, at least we would be able to say that under these controlled experimental conditions, different genes have different rates of mutation accumulation and they will be comparable to one another.

The first paragraph in the Discussion describes how the cell culture-related background signature reduces the signal-to-noise ratio. This could pose a challenge in detecting signatures. It is possible that some gene knockouts produce signals that are too weak to be detected under these experimental conditions.

The second paragraph in the Discussion describes whether increasing the level of mutagenesis could produce more mutations and therefore more signal. This could be achieved by increasing the level of damage in an acute (e.g. UV light) or chronic way (e.g. cisplatin treatment). However, the interpretation of the results needs to be done with this in mind. There are also alternative cell models which could be used but these sometimes have different genetic backgrounds and could generate diversity in signatures produced. However, again, the interpretation of mutation pattern outcomes needs to have this in mind.

We feel that we have covered these points in the Discussion and have chosen not to expand further given the already high word count.

2) In the case of those genes producing signatures, such as MSH6, the authors here have the opportunity to measure the ‘mutation rate’ of each signature per generation, however they do not seem to report it. This would be a very interesting measurement to see in the main manuscript.

This is a good suggestion. Our apology for not providing the mutation rate of each mutational signature in the previous manuscript. This has now been done in the revised manuscript.

In order to measure the mutation rate of each signature, we employed cell proliferation assays to determine the cell division rates for the nine knockout cell lines used in this study (Fig S1c). Based on the cell division rate and the exposure of each mutational signature, we calculated the mutation rate of the seven mutational signatures observed in this study. The average substitution rates of *MSH6* and *EXO1* knockout signatures are 148 and 16 per per genome cell cycle, respectively. The average indel rates of *MSH6*, *EXO1* and *FANCC* knockout signatures are 36, 0.58, 0.58 per genome cell cycle, respectively. The rearrangement rates of *EXO1* and *FANCC* knockout signatures are 0.19 and 0.68 per cell cycle, respectively (Supplementary Table 4). In the revised manuscript, we added a paragraph in the Online Methods on how the proliferation assay was performed, on page 11:

“Proliferation assay

Knockout cell lines were plated in triplicates at a density of 0.32×10^6 cells ml^{-1} and allowed to proliferate. Every second day, cells were dissociated with Trypsin-EDTA (Gibco), living cells were counted using CASY Cell Counter and Analyzer system (Innovatis), and replated at 1:2, 1:3 or 1:4 dilutions, depending on the growth rate of the cell line. The experiment was carried out for 10 days. Proliferation was plotted for each time point considering the dilution rates. The average growth rate is a mean over 10 days.”

We also added a paragraph to describe the mutation rate of the seven mutational signatures in the Results section, on page 7:

“To understand whether the targeting of these DNA repair genes could affect proliferation, we measured the proliferation rates of the given cell lines over a period of ten days (Supplementary Fig. 1c). The *MSH6*, *EXO1* and *FANCC* knockouts had the slowest proliferation rate, indicating that loss of these genes is not associated with an increased proliferative rate. Hence, the elevated numbers of mutations in *MSH6*, *EXO1* and *FANCC* knockouts were not simply due to an increase in the rate of cell division. Based on these assays, the mutation rates of the seven mutational signatures can be calculated: *MSH6* knockout signatures produced ~148 substitutions and ~36 indels per cell division; *EXO1*

knockout signatures produced ~16 substitutions, ~0.58 indels and ~0.19 rearrangements per cell division; *FANCC* knockout signatures produced ~0.58 indels and ~0.68 rearrangements per cell division (Supplementary Table 4).”

3) Figure 6 compares mutational signatures experimentally generated by the authors with breast cancer data. However, using circos plots to make this comparison renders the figure unreadable. It is impossible to determine the level of resemblance between the two datasets from this figure. I appreciate that the in vitro data cannot recapitulate perfectly what happens in humans, but at least a better way of comparing the actual signatures should be presented. Moreover, many more than just a handful of cases should be shown since large datasets of >500 WGS breast cancer samples are available to the authors.

We thank the reviewer for the suggestions.

We calculated cosine similarities between experimental-derived-KO signatures with cancer-derived signatures, and provided the plots in a new version of Fig. 6. Here, the substitution signatures of *MSH6* and *EXO1* knockouts were compared with 30 COSMIC signatures (<http://cancer.sanger.ac.uk/cosmic/signatures>). Indeed:

- The *MSH6* KO substitution signature has highest cosine similarity (0.91) to COSMIC Signature 20.
- Cosine similarities with other MMR-deficiency related signatures (COSMIC Signatures 6, 12, 14, 15, 20, 26) are all greater than 0.6.
- The *EXO1* KO substitution signature is most similar to cancer signature 3 (0.71) and 5 (0.71) – a note of caution in interpreting these values which are not like linear regression, they would not be considered to be very high, but they are certainly most alike to COSMIC Signature 3 and Cosmic Signature 5.

The rearrangement signatures of *EXO1* and *FANCC* knockouts were compared with six rearrangement signatures derived from 560 breast cancers⁵. The rearrangement signature of the *EXO1* knockout is very similar (0.93) to RS1 which is defined by long tandem duplications. By contrast, the rearrangement signature of the *FANCC* knockout shows low similarity to RS1 (0.09), but high similarity to RS3 (0.43) and RS5 (0.60), which have short tandem duplications and deletions.

For now, we have retained the genome plots of the experimental data as well as a small number of patient cancer samples, and we are happy to leave this with the editors to advise on whether to leave these whole genome plots in or otherwise. Although few examples are given, the direct comparisons between the genome plots of cancers and experimental knockouts provide an immediate visualization that the whole genome profiles of cancer and that of experimentally-generated signatures have enormous similarities. For example:

- The genome plot of two mismatch repair deficient samples and *MSH6* knockout all show overwhelming, high density of substitutions (dots) and indels (rings);
- The genome plots of two *BRCA1*-intact samples and *EXO1* knockout all show excessive long tandem duplications (inner green lines);
- The genome plots of two *BRCA1*-null samples and *FANCC* knockouts have mixed red and green lines in the center, indicating a mutational process generating both deletions and tandem duplications.

In addition to the change of Fig. 6 and its caption, we also revised the text considerably in this section on “**Striking similarities between experimentally-generated and cancer-derived mutational signatures**” which is noted in red in the manuscript on pages 7 and 8 and shown below.

“When mutational signatures were first mathematically extracted from cancers, several mutational signatures were found to be associated with inactivation of DNA repair genes. To investigate how *in vitro* experimentally-generated mutational signatures of gene knockouts compared with *in vivo* cancer-derived signatures, we calculated cosine similarities between the *in vivo* and *in vitro* mutational signatures for substitutions (Fig. 6a) and rearrangements (Fig. 6b) (cancer-derived indel signatures are not available). Then, we compared overall mutational profiles of knockouts with that of patient cancers.”

“The substitution signatures of *MSH6* and *EXO1* knockouts were compared with cancer-derived 30 COSMIC signatures (<http://cancer.sanger.ac.uk/cosmic/signatures>). The *MSH6* knockout signature is most similar to COSMIC signature 20 with cosine similarity of 0.91 (Fig. 6a), although there are relatively high cosine similarities when compared to other cancer-derived signatures associated with MMR-deficiency (all greater than 0.6). The *EXO1* knockout substitution signature is most similar to COSMIC Signatures 3 (0.71) and 5 (0.71).”

The reviewer also suggested showing more samples when making comparisons with between the knockout signatures and cancer samples. However, showing a large number of cancer profiles would probably not be a sensible thing for us to do.

Nevertheless, here, we have compared the mutational profiles of cancers that have been shown to be mismatch repair deficient in breast cancers based on immunohistochemistry of mismatch repair proteins, to the experimentally-generated mutation signature of *MSH6*. To reduce any ambiguity, we have separated the two cancer samples where mismatch repair deficiency (MMRd) was a late-onset, subclonal event (Apobec+MMRd) from the other two cohorts of MMRd and mismatch repair intact (non-MMRd) samples.

The figure below shows that cancers with MMRd have much greater similarity of substitution (Fig. 1a) and indel profiles (Fig. 1b) to the *MSH6* knockout signature, than non-MMRd cancers.

Figure 1. Cosine similarities between mutational profiles of cancer samples and mutational signatures of MSH6 knockout. Cancers were classified into three groups: Apobec+MMRd, MMRd and Non-MMRd. (a) Substitution. (b) Indels.

4) There seem to be a protein loading control missing in the western blot in the Supplementary Figure 1. This is essential to distinguish knockout effect from potential protein loading issue.

The different immunoblots were intended to function as loading controls for each other. However, we have now re-run these samples and include a loading control (Figure S1b).

5) Figure S8 d should be corrected to “c”

Our apologies. We thank the reviewer for pointing this out. It has been corrected.

Reviewer #3

1) The data produced here seems very interesting and I would make sure that the raw data is available post-publication.

Yes of course. We have created an ftp site for readers to download the mutation data directly. The ftp site will go live once the paper is published and will be situated here:
[/nfs/disk69/ftp/pub/cancer/Zou_et_al_2017](ftp://nfs/disk69/ftp/pub/cancer/Zou_et_al_2017)

We have included this in a “Data availability” section at the end of the Methods.

2) Regarding the genome editing aspects: The methods used to isolate clones and validate the knockouts are solid. I would be worried about acquiring additional mutations in the clonal expansion steps that would affect mutational signatures as well, e.g. additional mutations in DNA repair genes. Can the authors analyze the data to look for additional loss-of-function mutations in their clones?

In addition, there is no reference to sgRNA off target effects, to minimally address this issue the authors should computationally analyze off-targets of the sgRNAs chosen and show that there is no effect on other genes.

Thank you for the positive comments on the experimental set-up and analysis.

Indeed, the reviewer highlights an important point which is broadly relating to ensuring that we have considered appropriate negative controls. In other words:

- Have we excluded the possibility of off-target effects?
- Have we excluded the possibility of a deleterious mutation acquired in the clonal expansion phase, that could contribute to the downstream mutational signatures of subclones? Mutations that arise after single-cell bottlenecking or are acquired during the earliest phase of clonal expansion could affect a cell that is picked to be sequenced. If so, we should be able to see this if we were to explore the coding sequences of the subclones.

Thus, we performed a genome-wide assessment looking for off-target effects that could hit

- 142 DNA repair genes in all the parental clones and all the subclones
- Any other gene in all the parental clones

We did not find additional deleterious, coding mutations in these genes. We therefore surmise that the mutational signatures of knockouts observed in subclones were indeed induced purely through knocking out specific target-genes.

Furthermore, we used a web-based tool COSMID (<http://crispr.bme.gatech.edu>) to search for potential off-target sites of nine chosen guide RNA sequences. According to the given guide RNA sequence, COSMID generated a ranked list of perfectly matched and partially matched sites (these sites can be on-target sites or possibly other sites) in the genome as well as their ranking score. According to the ranking score and the position of other sites, we could scrutinize in more detail, predicted off-target effects of each guide RNA sequence. From the results, we can confirm that the off-target effect was minimal and there was no effect on other genes. The results generated by COSMID is provided in Supplementary Table 2. We have made the following modifications in red on page 4 of the current manuscript:

“Potential off-target edits were also systematically sought in an agnostic manner, whether generating small or large (multi-kb) insertion or deletions, and none were identified. Proliferation rates were also determined for each knockout cell line (Supplementary Fig. 1c). Moreover, potential off-target sites were also searched using COSMID (<http://crispr.bme.gatech.edu>), a web-based tool to identify and validate CRISPR/CAS9 off-target sites⁶ (see Supplementary Table 2 for a ranked-list of potential off-target sites of the relevant guide RNA sequences generated by COSMID). Furthermore, we also confirmed in all subclones, that no additional mutations were acquired in other DNA repair genes during the early clonal expansion phase (see Supplementary Table 3 for a list of DNA repair genes) that could affect the final mutational signature obtained in each subclone”

We thank the reviewer for highlighting this important point.

3) There is minimal intergenic off-targets, as a large number of intergenic off-target effects can cause elevated DNA damage that will affect cell viability and maybe also the DNA damage machinery. To maximally address this issue the authors can show that two sgRNAs that target the same genes (e.g. on two different exons) produce the same result.

The reviewer points out that a large degree of intergenic off-target edits could affect cell viability by introducing an excess of DNA damage. However, we have performed both systematic, agnostic searches as well as an informed searched for potential off-target sites using COSMID, and we do not find a high level of systematic off-target mutations.

To address the reviewer’s other concern in this point, and show that an alternative sgRNA would produce a similar mutational signature effect, we have used two alternative guide RNA sequences for *MSH6*, *POLB* and *POLM* and we have performed these experiments in a different cell-based system - induced pluripotent stem cells. Unlike HAP1 cells, these are not cancer cells and they are pluripotent. We still observed the same result as in HAP1 experiments:

- *MSH6* knockouts showed a near identical mutational signature as observed in HAP1 cell line

- Whilst *POLB* knockouts and *POLM* knockouts remained demonstrating a strong background signature and no specific signature of their own (Fig. 2).

Figure 2. *MSH6*, *POLB*, *POLM* knockouts in a human iPS cell line (Unpublished data) show the similar mutational patterns to the gene knockouts in HAP1 cell line.

4) Another issue that need to be referenced is the choice of clonal expansion vs. polyclonal experiments. Using lentiviruses to deliver Cas9 and sgRNAs can result in a close to complete KO levels in a polyclonal population. Would such an approach simplify or complicate the experiments presented in this paper?

We designed these experiments to be as controlled as possible. We feel that there are risks with leaving the parental population as polyclonal post-delivery of sgRNAs.

- First, even if the efficiency of editing was 100%, there is a risk that some edited genotypes would not result in a deleterious frameshifting mutation.
- Second, the risk with performing the mutation accumulation step as a polyclonal population is that when it comes to the final step (the single-cell bottleneck step for the subclones - this must be done because otherwise, we cannot “see” the mutations – they need to be clonal), we risk picking out cells that then have not been knocked out, e.g., they carry a loss of 3bp resulting in an in-frame deletion.
- Furthermore, each of the subclones will have a different genotype. If there were some variation between subclones, it would be difficult to say whether that was due to fluctuation in distribution or due to the genotype.

In all, we feel that our version of experimental set-up may be conservative, and longer, but is well-controlled.

5) Lastly, does any of these gene KOs affect growth rate which would affect the number of cell divisions and the quantitative mutational outcome?

We have now calculated the proliferation rate of all cell lines used within our study. Interestingly, the knockouts showing the clearest mutation signatures are not the ones that

proliferate fastest, hence, we can rule out the argument that knockouts that did not show a signature were simply due to not having been allowed to proliferate as much.

This question is related to the question 2 asked by Reviewer #2, so please see our response above for a more detailed answer.

Reviewer #4:

1) Regarding the MSH6 knockout: The authors claim that the mutational signature “bears a resemblance to the multiple substitution signatures that have been associated in cancers, but was not perfectly identical to any of them.” Why do they compare the mutational signatures to “cancers” in general?

We make a comparison to cancers because we expect this to be a natural question from readers, given that mutational signatures were first ascertained in cancers and we are trying to validate this purely mathematical concept through these experiments.

2) How did it compare to colorectal or endometrium carcinomas, the two tumor entities associated with MSH6 germline mutations?

Although *MSH6* germline mutations are associated with an increased risk of developing colorectal and uterine cancers, the incidence of *MSH6* germline mutations amongst colorectal and endometrial cancers is infrequent^{7,8}. Indeed, in a recent study on Pan-Cancer Analysis of Whole Genomes⁹ (PCAWG) germline variants from 2642 cancers of all types, there were no tumours that had a germline *MSH6* mutation with loss of the wild-type parental allele (i.e. biallelic *MSH6* loss)¹⁰. There are three samples with somatic mutations in *MSH6* (likely to be a single hit on *MSH6* in each sample), all of them were *POLE*-mutated as well, which complicates signature assessment. In addition, we also checked colorectal and endometrial carcinomas in TCGA, and did not identify any samples with genetic biallelic loss of *MSH6*.

The thirty mutational signatures on COSMIC (<http://cancer.sanger.ac.uk/cosmic/signatures>) were derived from a *pooled analysis of 40 distinct types of human cancer-types* (performed by a different scientific group). A number of those signatures are associated with MMR deficiency (Signatures 6, 12, 14, 15, 20 and 26).

In our manuscript, we have compared our *MSH6* knockout signature with the COSMIC signatures because these are generally seen to be the gold-standard. We obtain great similarities between our experimentally-generated signature and those on COSMIC but they are not perfectly identical. We believe that this is likely to be due to how the signature analysis was performed on COSMIC - where data across many tumor-types have been pooled, in numbers that are not equivalent between different tumor-types.

When we perform signature extraction on individual tissue-types alone, (i.e. breast cancer specifically, the 52 colorectal cancers and 44 uterine whole-genome sequenced cancers available on ICGC specifically) we in fact obtain a MMR-deficiency mutational signature that is more similar to our experimental data than the COSMIC signatures are. In response to the reviewer’s question, we added the text highlighted in red on page 6:

“Interestingly, when mutational signatures are extracted from breast cancers alone and all analyses restricted to just this tissue-type, we find that the *in vitro* signature is strikingly similar to the MMR deficiency signature in breast cancers. **This is also the case for tumor-specific signature extractions of 52 colorectal and 44 endometrial cancers, both being cancer-types that are associated with *MSH6* mutations.**”

3) They describe a striking similarity to the MMR deficiency signature in breast cancer. However, the same group has published recently a paper (Cancer Res 77:4755-62), where only 11 of 640 breast cancers were found to be MMR deficient. Apparently, none of them had an inactivation mutation in *MSH6*; in fact, inactivation mutations in MMR-related genes were found in only six cases. Given the sparse available data on MMR-deficient breast cancers, what data set is the authors’ claim of similarity to biological cancers based on?

MMR deficiency is not common in breast cancer. This is why they are not looked for when patients are diagnosed with breast cancer. MMR deficiency is essentially never detected. Our manuscript in Cancer Res was intended to highlight how taking a whole genome sequencing and mutational signatures approach allowed us to detect 11 patients with clear MMR deficiency that would otherwise have simply been missed.

The reviewer is correct that none of the 11 had clear genetic inactivation of *MSH6*. In fact, we were unable to confirm the genetic inactivation of any of the mismatch repair genes in 5 of the 11 patients. However, we performed an alternative, independent test, that is protein immunohistochemistry (IHC), to show that there were concomitant losses of *MSH2/MSH6* and *MLH1/PMS2* even in the patients where we could not find the genetic defect. In other words, the mutational signatures were clearly correlating with the protein IHC result, even when we could not find the causative gene defect. The confirmation by IHC is the basis of our comparison.

Perhaps there are alternative ways of inactivating components of mismatch repair that we do not fully understand. There is also the possibility that other closely-related genes are influencing the mismatch repair pathways and producing very similar outcomes to knocking out these particularly well-known genes.

4) The authors describe similarities between experimentally generated and cancer-derived mutational signatures as “striking” or “uncanny”. However, to this end, Figure 6 is not convincing. Firstly, the scale of the figure is not suited to identify all the differently colored dots. Secondly, there seem to be clear differences between the plots. For example, in Fig. 6d in the outermost circle blue is the dominating color, which is not the case in the corresponding Figs. 6e-f. Similarly, the colors in the second outer circles do not match and neither do the rearrangements in the innermost circles. The same applies also to Figs. 6a-c and Figs. 6g-i. In order to provide convincing evidence for the claim of “striking similarities”, either some robust statistics have to be presented or the figure has to be changed. At present, it remains unclear whether the *in vitro* data have any relevance for the *in vivo* scenario.

We thank the reviewer for these thoughts. The second reviewer also suggested improvements in this comparison between *in vitro* and *in vivo* data. Please the detailed answer to question 3 from Review 2 and the associated modifications to figure and text.

5) The observation that the same single gene defect in isogenic cell lines under standardized culture conditions can result in different mutational signatures is intriguing and not easy to grasp. Unfortunately, the authors do not discuss this, but they should provide their views how this can be explained.

We thank the reviewer for highlighting this oversight. We have added the following sentence to the text on page 7.

“Furthermore, single gene targeting *in vitro* in some cases generated not just one but multiple mutational signatures, buttressing previous reports that multiple *in vivo* cancer-derived signatures could arise from single gene defects such as in *BRCA1/BRCA2*¹¹. **This is likely to be due to the multitude of compensatory DSB repair pathways that are brought into play in the absence of conservative, error-free HR and due to some activity of translesion synthesis. Whatever are the mechanisms that underpin these observations, this is important authentication – because multiple mutational signatures are now starting to be exploited as a principle for designing clinical biomarker assays¹¹.**”

1. Welch, P.L. & King, M.-C. BRCA1 and BRCA2 and the genetics of breast and ovarian cancer. *Human Molecular Genetics* **10**, 705-713 (2001).
2. Zheng, L., Li, S., Boyer, T.G. & Lee, W.-H. Lessons learned from BRCA1 and BRCA2. *Oncogene* **19**, 6159 (2000).
3. Niedzwiedz, W. *et al.* The Fanconi Anaemia Gene FANCC Promotes Homologous Recombination and Error-Prone DNA Repair. *Molecular Cell* **15**, 607-620 (2004).
4. Garcia-Higuera, I. *et al.* Interaction of the Fanconi Anemia Proteins and BRCA1 in a Common Pathway. *Molecular Cell* **7**, 249-262 (2001).
5. Nik-Zainal, S. *et al.* Landscape of somatic mutations in 560 breast cancer whole-genome sequences. *Nature* **534**, 47--54 (2016).
6. Cradick, T.J., Qiu, P., Lee, C.M., Fine, E.J. & Bao, G. COSMID: A Web-based Tool for Identifying and Validating CRISPR/Cas Off-target Sites. *Molecular Therapy. Nucleic Acids* **3**, e214 (2014).
7. Wijnen, J. *et al.* Familial endometrial cancer in female carriers of MSH6 germline mutations. *Nature Genetics* **23**, 142 (1999).
8. Peterlongo, P. *et al.* MSH6 germline mutations are rare in colorectal cancer families. *International Journal of Cancer* **107**, 571-579 (2003).
9. Campbell, P.J., Getz, G., Stuart, J.M., Korbil, J.O. & Stein, L.D. Pan-cancer analysis of whole genomes. *bioRxiv* (2017).
10. Waszak, S.M. *et al.* Germline determinants of the somatic mutation landscape in 2,642 cancer genomes. *bioRxiv* (2017).
11. Davies, H. *et al.* HRDetect is a predictor of BRCA1 and BRCA2 deficiency based on mutational signatures. *Nat Med* **23**, 517-525 (2017).

REVIEWERS' COMMENTS:

Reviewer #1 (Remarks to the Author):

The authors have addressed all of the concerns raised by me.
The manuscript has improved.

Reviewer #2 (Remarks to the Author):

The authors addressed all my comments.

Reviewer #3 (Remarks to the Author):

The authors have addressed my concerns

Reviewer #4 (Remarks to the Author):

The authors have addressed all the comments and criticisms I have raised in my review of the original manuscript.